# Liver Steatosis, Gut-Liver Axis, Microbiome and Environmental Factors. A Never-Ending Bidirectional Cross-Talk

**DOI:** 10.3390/jcm9082648

**Published:** 2020-08-14

**Authors:** Agostino Di Ciaula, Jacek Baj, Gabriella Garruti, Giuseppe Celano, Maria De Angelis, Helen H. Wang, Domenica Maria Di Palo, Leonilde Bonfrate, David Q-H Wang, Piero Portincasa

**Affiliations:** 1Clinica Medica “A. Murri”, Department of Biomedical Sciences and Human Oncology, University of Bari Medical School, 70124 Bari, Italy; agostinodiciaula@tiscali.it (A.D.C.); domenicamariadipalo@gmail.com (D.M.D.P.); leonildebnf@gmail.com (L.B.); 2Department of Anatomy, Medical University of Lublin, 20-090 Lublin, Poland; jacek.baj@me.com; 3Section of Endocrinology, Department of Emergency and Organ Transplantations, University of Bari “Aldo Moro” Medical School, Piazza G. Cesare 11, 70124 Bari, Italy; gabriella.garruti@uniba.it; 4Dipartimento di Scienze del Suolo, della Pianta e Degli Alimenti, Università degli Studi di Bari Aldo Moro, 70124 Bari, Italy; g.celano1@gmail.com (G.C.); maria.deangelis@uniba.it (M.D.A.); 5Department of Medicine and Genetics, Division of Gastroenterology and Liver Diseases, Marion Bessin Liver Research Center, Einstein-Mount Sinai Diabetes Research Center, Albert Einstein College of Medicine, Bronx, NY 10461, USA; helen.wang@einsteinmed.org (H.H.W.); david.wang@einsteinmed.org (D.Q.-H.W.)

**Keywords:** body weight, fat, fatty liver, genetics, intestinal permeability, leaky gut, NAFLD

## Abstract

The prevalence of non-alcoholic fatty liver disease (NAFLD) is increasing worldwide and parallels comorbidities such as obesity, metabolic syndrome, dyslipidemia, and diabetes. Recent studies describe the presence of NAFLD in non-obese individuals, with mechanisms partially independent from excessive caloric intake. Increasing evidences, in particular, point towards a close interaction between dietary and environmental factors (including food contaminants), gut, blood flow, and liver metabolism, with pathways involving intestinal permeability, the composition of gut microbiota, bacterial products, immunity, local, and systemic inflammation. These factors play a critical role in the maintenance of intestinal, liver, and metabolic homeostasis. An anomalous or imbalanced gut microbial composition may favor an increased intestinal permeability, predisposing to portal translocation of microorganisms, microbial products, and cell wall components. These components form microbial-associated molecular patterns (MAMPs) or pathogen-associated molecular patterns (PAMPs), with potentials to interact in the intestine lamina propria enriched in immune cells, and in the liver at the level of the immune cells, i.e., Kupffer cells and stellate cells. The resulting inflammatory environment ultimately leads to liver fibrosis with potentials to progression towards necrotic and fibrotic changes, cirrhosis. and hepatocellular carcinoma. By contrast, measures able to modulate the composition of gut microbiota and to preserve gut vascular barrier might prevent or reverse NAFLD.

## 1. Introduction

Non-alcoholic fatty liver disease (NAFLD) is the most common cause of chronic liver disease worldwide [1].

The acronym NAFLD refers to the presence of hepatic steatosis without other causes for secondary hepatic fat accumulation. By definition, NAFLD subjects have no signs of acute or chronic liver disease other than fatty liver and consume no or little alcohol [2] (Table 1).

In the USA, NAFLD occurs in more than 65 million residents and causes enormous economic burden, i.e., >100 billion USD annually [3]. NAFLD worldwide has a prevalence of almost 30–40% in the adult population and increases in type 2 diabetes mellitus (50%) [1,4,5], obesity (30–76%) [6], and morbid obesity (up to 90%) [7,8].

Early identification and targeted treatment of patients with NAFLD can prevent consequences related to complications (including management of end-stage disease and HCC) and rising health cost. Further beneficial effects can derive the reduction of risk factors for extrahepatic complications, which include cardiovascular disease and malignancy [9,10].

This review discusses the most recent updates on the gut–liver axis and the aspects involving the gut barrier, intestinal permeability, the microbioma, and the role of local molecules and environmental factors with respect to emerging problem of NAFLD worldwide (Figure 1).

The liver secretes primary bile acids and antimicrobial molecules (immunoglobulin A (IgA) and angiogenin) into the biliary tract. Molecules reach the intestinal lumen and contribute to maintenance of gut eubiosis (while inhibiting intestinal bacterial overgrowth). During the enterohepatic circulation of bile, bile acids (BAs) act as signaling molecules by interacting with the nuclear receptor farnesoid X receptor (FXR) and the G protein-coupled bile acid receptor (GPBAR1). This interaction contributes to modulate hepatic bile acid synthesis, glucose metabolism, lipid metabolism, and dietary energy utilization. Gut products include host and/or microbial metabolites and microbial-associated molecular patterns (MAMPs). The systemic circulation also plays a role, since products are translocated to the liver via the portal vein and can influence liver functions. The systemic circulation extends the gut–liver axis since liver metabolites from dietary, endogenous, or xenobiotic substances (i.e., free fatty acids, choline metabolites, and ethanol metabolites) are transported to the intestine through the capillary system. This mechanism of continuous recirculation of molecules across blood capillaries can distinctively affect the intestinal barrier. Butyrate, the short chain fatty acid, is protective (i.e., improves the colonic defensive border), while acetaldehyde can activate a barrier damage.

## 2. Risk Factors, Clinical Features, and Pathophysiology of NAFLD

### 2.1. Definition of NAFLD

The term NAFLD comprises a wide spectrum of conditions [13,14], ranging from simple non-alcoholic fatty liver (NAFL) without significant inflammation (the majority of cases), to non-alcoholic steatohepatitis (NASH) (affecting about 5% of the population). In NASH, hepatic steatosis is associated with pericellular fibrosis, ballooning degeneration of hepatocytes, lobular inflammation, and apoptosis [15]. NASH has potential to progress to (cryptogenic) cirrhosis in 20% of cases. Furthermore, hepatocellular carcinoma (HCC) can originate from cirrhosis [6,16,17,18] but also from NASH [19].

### 2.2. Risk Factors

NAFLD can be considered as the hepatic expression of gluco-lipid metabolic disturbances that includes insulin resistance and type 2 diabetes mellitus, expansion of visceral fat/frank obesity, dyslipidemia (involving hypertriglyceridemia and low HDL-cholesterol), and ultimately metabolic syndrome (MetS). In particular, triglyceride/HDL-cholesterol ratio [20] and total cholesterol/HDL-cholesterol ratio [21] are independently linked with the presence of NAFLD, being considered reliable predictors of liver disease.

The term NAFLD has recently received a different acronym, i.e., MAFLD (metabolic-dysfunction associated fatty liver disease). According to this novel definition, MAFLD consists of “inclusive” (not only exclusive) criteria relying on evidence of hepatic steatosis, in addition to one of the following three criteria, namely overweight/obesity, presence of type 2 diabetes mellitus, or evidence of metabolic dysregulation [22]. Indeed, body mass index (BMI) correlates with the increasing prevalence of both NAFLD and NASH [23,24]. The cut-off values for overweight and obesity, however, differ with ethnicity. In the Caucasian race, overweight and obesity occur at a BMI 25.0–29.9 kg/m^2^ and ≥30 kg/m^2^, respectively. In the Asian population, overweight and obesity occur at a BMI 22.5–24.9 kg/m^2^ and ≥25 kg/m^2^, respectively. Furthermore, Asia has the most worrisome trends with BMI when stratified by ages [25]. NAFLD can be a feature in obese subjects without metabolic abnormalities or even nonobese subjects (body mass index (BMI) < 25 kg/m^2^) with insulin resistance [26,27].

Thus, NAFLD can also develop in lean (likely metabolically-impaired) subjects as well [28] and this possibility is frequent in Asia [29]. Besides genetic factors, the onset and progression of NAFLD in lean subjects is related with dietary and environmental factors, and show metabolic characteristics (i.e., insulin-resistance, dyslipidemia) similar to those observed in overweight and obese subjects [30].

Several lean individuals in the United States suffer from NAFLD, possibly due to the coexistence of diabetes and/or hypertension [31]. In a large cross-sectional Asian study, the presence of NAFLD or AFLD was associated in non-obese subjects (BMI < 25 kg/m^2^) with the score of coronary artery calcification, an expression of subclinical atherosclerosis [32]. Additionally, NAFLD prevalence and liver fibrosis has been found to increase with age [33].

### 2.3. Clinical Aspects

While most subjects with NAFLD remain asymptomatic, some patients, especially those developing NASH, may complain of vague symptoms (i.e., fatigue, malaise, and right upper abdominal discomfort). NAFLD subjects could ask for medical advice because of mildly elevated liver alanine aminotransferases or incidental increased echogenicity (“bright liver”) at ultrasound (or eventually decreased hepatic attenuation by computed tomography, as well as increased fat signal by magnetic resonance imaging). NAFLD, however, may develop with normal aminotransferase levels. The ultimate diagnosis of NAFLD requires the confirmation of steatosis by liver histology or imaging, no relevant alcohol consumption. or other causes of liver steatosis, and exclusion of a coexisting other chronic liver disease.

The early fat storage in the liver could drive subclinical liver abnormalities, leading to NAFLD development and progression, that can increase risk for advanced liver disease and liver-related mortality [33]. NAFLD also increases the risk of non-liver-related complications, such as cardiovascular disease and malignancy [2,9,10,34,35,36]. In addition, patients with NAFLD suffer from poor quality of life as compared to healthy individuals. A marked worsening occurs with advanced liver diseases, such as liver cirrhosis [37,38]. No specific therapy exists for NAFLD, apart from healthy lifestyles [28,39]. The utility of systematic screening of the population is still debated because of unclear cost-effectiveness [40]. Specific research protocols can target subgroups of patients with symptomatic or asymptomatic NAFLD [41].

### 2.4. Pathophysiology of NAFLD

The pathophysiology of NAFLD is complex, multifactorial, and partly unknown [28,42,43].

NAFLD is due to excessive (>5%) storage of hepatic triglycerides (TG) as micro- and macro-vesicular deposits, associated with an accumulation of free fatty acids (FFAs), ceramides, and cholesterol [44,45]. In health, as shown in Figure 2, there are three essential steps of FFA accumulation in the liver:(a)Influx of non-esterified fatty acids (NEFA): This step accounts for ~60% of total FFAs in the liver and brings to enrichment of the FFA pool in hepatocytes. NEFA originates from lipolysis of triglycerides in adipocyte under the control of insulin [46]. Once in the liver, FFAs have different fates:As fatty acyl-CoA, FFAs can enter the mitochondria under the control of carnitine palmitoyltransferase (CPT)-1 and undergo β-oxidation to acetyl-CoA joining the tricarbossilic acid cycle with production of ATP;FFAs are also esterified to TG (no more than 5% in the hepatocyte) via the key enzymes diaglyceride acyltransferase (DGAT)1 and DGAT2. This step is also controlled by insulin and excess TG can be stored as lipid droplets or exported to blood as very-low-density lipoprotein [47].TG can also be hydrolyzed under the actions of hydrolases such as the Patatin-like phospholipase domain-containing protein 3 (PNPLA3), also known as adiponutrin, to enrich the fatty acid (FA) pool [46,48].(b)De novo lipogenesis (DNL) of FAs: This step accounts for ~25% of total FFAs in the liver and originate from dietary sugars. With DNL, hepatocytes converts excess glucose and fructose to FAs. Insulin mediates the transport of absorbed dietary carbohydrates to its target tissues (skeletal muscle and liver for storage as glycogen). Glucose in hepatocytes is partly metabolized to pyruvic acid via glycolysis and then to acetyl-CoA, to generate ATP in the tricarboxylic acid cycle and promote gluconeogenesis during hypoglycemia. The first step of DNL is the synthesis of malonyl-CoA from cytosolic acetyl-CoA by acetyl-CoA carboxylase (ACC). Malonyl-CoA serves as a substrate to form saturated FA via FA synthase (FAS). Stearoyl CoA desaturase (SCD)1 is an endoplasmic reticulum (ER) enzyme responsible for the formation of monounsaturated FAs from saturated FAs [41].(c)Influx of dietary FAs: Influx of FAs from diet is ~15% of all amount of FFAs in the liver [48]. Bile acids (BA) hydrolyze intestinal dietary TGs to form nascent chylomicrons. The FAs made from TG hydrolysis are taken up by adipose tissue and liver [49]. At the same time, BA acts as a potent metabolic regulator in the terminal ileum by activating the farnesoid X receptor (FXR) plus the pregnane X receptor (PXR), as well as the G-protein-coupled bile acid receptor-1 (GPBAR-1). These interactions produce effects in the liver and in the muscle, in adipocytes and brown adipose tissue for energy expenditure [12,50].

Metabolic abnormalities can markedly disrupt the pathways governing FFA/TG/lipid metabolism in the liver, leading to NAFLD and damage of hepatocyte (Figure 3 and Figure 4). The progressive development of insulin resistance, expansion of visceral obesity, sedentary behaviors, and high-calorie diet can all affect FFA homeostasis. This ongoing inflammation induced by metabolic stress is associated with an increase of lipolysis and influx of NEFA, de-novo lipogenesis, influx of dietary FFA (especially fructose), TG deposition in droplets, and the decrease of FFA in mitochondrial oxidation, or secretion/export of FFA or TG into VLDL [51]. Ingested glucose is re-directed to the liver during insulin resistant status [47] and glucose is primarily processed to FA via DNL [46,52]. If FFAs synthesis increases, they are converted to fatty acyl-CoA, which is further esterified into TGs and stored in hepatocytes. DNL is also enhanced in NAFLD patients [47,53]. In general, the oversupply or disrupted disposal free fatty acids (FFAs) in the liver activates mechanisms that leads to the assembly of lipotoxic species, i.e., FFAs, TGs, Lysophosphatidylcholine (LPC), ceramides, and free cholesterol (Table 2).

Notably, the onset and progression of NAFLD does not seem to be exclusively secondary to an exaggerated caloric intake. In particular, loaded cholesterol in the hepatocyte can impair mitochondrial and lysosomal function, with direct toxic effects (i.e., oxidative stress and lipotoxicity) in both nonobese and obese patients [60]. Free cholesterol and oxLDL can also accumulate in the portal vein wall and this step can predispose to portal venous NLRP3 inflammasome-mediated inflammation and fibrosis leading to NAFLD [61].

Furthermore, the onset and progression of NAFLD can also be linked with exposure to environmental chemicals (e.g., volatile organic chemicals, persistent organic pollutants, metals, particulate matter, pesticides) through several mechanisms disrupting biochemical, metabolic, signaling, and endocrine pathways [62].

In general, the mechanisms of lipotoxicity involved in NAFLD onset and progression are multiple and involve receptor/kinases-mediated interactions, the endoplasmic reticulum/other intracellular organelles, mitochondria [63], the nucleus, and several signaling pathways. Endoplasmic reticulum stress, inflammatory changes (metabolic inflammation or meta-inflammation) [64], production of reactive oxygen species, and cell death [65] represent further events associated with liver damage. The innate immune system contributes to metabolic inflammation with the recruitment of Kupffer cells, dendritic cells, lymphocytes, as well as hepatocytes and endothelial cells [66,67].

Ultimately, the activation of several transcription factors promotes the release of inflammatory cytokines and chemokines, and stimulates hepatic stellate cells with propensity to collagen deposition, and further aggravation of the insulin resistance status, both potential harmful features in the progression of NASH [68]. In addition, the adipose tissue, skeletal muscle, the heart, the pancreatic islets, certain areas of the brain (mainly hippocampus, cerebellum, hypothalamus), and the intestinal microbiota represent additional organs potential targets of lipotoxicity [41,69,70,71].

## 3. Beyond NAFLD: The Gut Liver-Axis, the Gut Barrier, and the Liver Barrier

NAFLD develops because of the interaction of genes (epistasis) and environmental factors (exposome) [74,75]. The environmental factors act through intestinal, microbial, and dietary modifications [11,76], and can be linked with exposure to food contaminants, contaminated consumer products or air pollution [77,78,79,80,81,82,83,84].

Changes in this subtle equilibrium involving networking, bidirectional cross-talk, and control of inflammation paves the way to initiation, perpetuation, and aggravation of metabolic liver damage [85]. For example, a diet enriched in fat could easily change the intestinal mucus [86] and the microbiome, leading to alterations in the intestinal barrier and gut vascular barrier, therefore promoting the influx of bacterial products in the portal tract towards the liver [87] with consequences on local and systemic inflammation controlling metabolic abnormalities.

The term gut-liver axis refers to the bidirectional relationship between the intestinal microbiome, the gut, and the liver [88]. There is a close communication between the liver and the intestine, which also involves the biliary tract, the portal vein, the systemic circulation, and a series of systemic mediators [11]. The interaction is bidirectional since gut-derived products permeate the intestine going through the portal vein to the liver, while liver-secreted bile and antibodies are flowing from the liver through the intestine. Moreover, BAs are markedly influenced by the intestinal microbiome but also control the intestinal microbiome, besides their classical physiologic functions that allow food digestion and absorption, as well as stimulation of nuclear receptors and specific receptors at the intestinal level [12,89,90]. The gut barrier integrity results from a balanced cross-talk between microbiome, mucus, intestinal cells, immunological system and gut-vascular barrier. The stability of this equilibrium is essential in maintaining intestinal homeostasis [91,92,93] (Figure 5).

### 3.1. Intestinal Microbiome

A first level of the gut barrier is the resident microbiota, which is composed by hundred trillions of microorganisms [94], with more than 10 times the genes of the human genome and a weight of about 1–2 kg [95]. The gut microbiota plays a key role in nutrient and vitamin processing, and biotransformation of intestinal primary BAs to secondary BAs [12,89,96].

### 3.2. Intestinal Mucus

A second level of gut barrier is the intestinal mucus, i.e., heavily glycosylated proteins secreted by the intestinal goblet cells [97]. This barrier represents an extracellular layer. The thickness of the mucus increases from the stomach towards the colon [98,99]. An important function of the mucus is the prevention of direct contact between harmful/toxic substances/microbiome with the enterocyte. The microbiota in health is physically separated from the intestinal epithelium by the mucus. The microbiota colonizes the outer layer of the mucus and microorganisms use nutrients from the mucus itself.

In general, the microbiota is able to cross-talk with the host via several metabolic products. Secondary metabolites include short chain fatty acids (SCFAs) like propionate, butyrate, acetate (produced during dietary fiber degradation), vitamins, and immunomodulatory peptides [100,101,102,103,104].

The outer layer is populated by bacteria attached directly or via mucin-IgA interaction (IgA produced by plasma cells and transported transcellularly in the enterocytes) and prevents the washing effects of peristalsis [105]. Paneth cells also dismiss antibacterial lectins, such as the regenerating islet-derived protein III (REG3G) that inhibits bacterial adhesion to the mucosa. While intestinal mucin controls the layering of the microbiota in the lumen, the microbiota itself contributes to shape the mucin [106], likely via signaling to goblet cells and the pathway involving the inflammasome NLRP6 [107]. Bacteria also stimulates cell-mediated immunity via Toll-like receptor (TLR)-mediated signaling [108]. In addition, some bacteria use mucus for nutrition and modulate inflammatory changes. In particular, *Akkermansia muciniphila* is a mucus degrading bacteria and its abundance is higher close to the mucus layer [109]. This anaerobe, Gram negative, mucus degrading specialist populates the intestinal lumen [110,111] and its reduced abundance is associated with inflammation, impaired barrier integrity, and non-alcoholic liver damage [112,113]. Mucin-degrading bacteria increase and mucus thickness decreases in the absence of dietary fiber [114]. Mucin glycosylation is also under the control of the ratio Bacteroides:Firmicutes [115]. Secondary metabolites are able to modulate other function, namely the differentiation of immune cells, i.e., T regulatory cells [116], macrophages, and microbicidal activity [117]. Effects of secondary metabolites are possible upon fiber metabolization, which involve white and brown adipose ratio [118].

Notably, the inner mucus layer is sterile because it does not harvest bacteria due to enrichment in in antimicrobial peptides and in proteins excluding bacteria (lypd8 and zymogen granulae protein 16, ZG16) [119] is rather static (unstirred), and is in contact with epithelial cells. This level contributes to the absorption of water and nutrients [98].

In summary, the mucus is a dynamic structure conferring protection to the host [98,120]. Changes of mucus and diet have consequences on microbiota distribution and composition. In ulcerative colitis microorganism come in contact with the epithelium and can perpetuate the local inflammation [121]. If the mucus function fails and qualitative/quantitative changes of mucus occur, inflammation is possible with absorption of toxic substances, as seen in cystic inflammatory bowel disease (IBD) and cystic fibrosis. Mice models show that a high MUC2 mucin production increases the susceptibility of goblet cells to apoptosis and endoplasmic reticulum stress, while alcohol intake and cirrhosis is associated with increased mucus thickness. In mice, abnormal MUC2 inside the epithelial cells leads to inflammatory changes, which resemble the ulcerative colitis. In addition, high-fat diets can disrupt the intrinsic structure of colonic mucin, as evident in mice developing liver steatosis [86,122].

### 3.3. Gastrointestinal Motility, Secretions, and Enterohepatic Circulation of BAs

A third level of the gut barrier is a dynamic assembly. It depends on the kinetics of gastrointestinal motility and secretions, with both events influencing the outer part of the mucus layer. This situation prevents the proliferations of microorganism and provides clearance of luminal debris, contributing to protection against pathogens. Fundamental fluids are the gastric acid and bile containing BAs as one of the three species of biliary lipids (together with cholesterol and phospholipids) [12]. Both fluids have antimicrobial properties [91]. In the stomach and small intestine, only *Helicobacter pylori* and *Lactobacilli* respectively survive in the acidic environment [123]. Change of these conditions might lead to both qualitative and quantitative modifications of the gut microbiota composition, abnormal intestinal homeostasis, and disease [91].

The enterohepatic circulation of bile and BAs plays a key role at the level of the gut-liver axis and the intestinal microbiota is in close, bidirectional contact with BAs [124,125,126,127,128]. The maintenance of the body BA pool depends on hepatic BA synthesis, biliary secretion, gallbladder concentration and contraction, intestinal transit, microbial biotransformation, intestinal re-absorption, and fecal excretion. In the liver the primary BAs (cholic acid (Ca) and chenodeoxycholic acid (CDCA)) are synthesized from cholesterol within the “classical pathway” by the rate-limiting microsomal enzyme cholesterol 7α-hydroxylase (CYP7A1) and CYP8B1 at a later step. Within the “alternative pathway”, the CYP27A1 enzyme is involved in BAs synthesis. BAs are conjugated to the amino acids glycine or taurine by the enzymes BA CoA synthase (BACS) and BA-CoA-amino acid N-acetyltransferase (BAAT). This process increases the solubility of BAs for secretion into bile by the bile salt export pump (BSEP) [129]. During fasting, most of the bile flows from the liver and is concentrated into the gallbladder, which acts as a reservoir. Gallbladder contraction in the fasting state secretes bile periodically into the duodenum where the intestinal circulation starts. About 20% emptying takes place in the fasting state at the end of phase II of the migrating myoelectric complex [130,131] under the control of the vagus and enterohormone motilin [131]. More than 50–60% gallbladder emptying occurs after a meal due to the enterohormone cholecystokinin (CCK) [132]. Periodic episodes of gallbladder relaxation/refilling several times during the day, accounting for the rhythmic activity of the gallbladder and pulsatile secretion of BAs in the intestine. This function requires the inhibitory effect of mediators on the gallbladder like the vasointestinal peptide (VIP, released in the duodenum by gastric acid) [133], BAs (via the gallbladder receptor GPBAR-1) [12,127], and the intestinal FGF15/19 (following the BA/FXR interaction in the ileum) acting on the FGF4/β-Klotho receptor also expressed in the gallbladder [132,134,135].

Most of the BAs (>95%) undergo terminal ileal active reabsorption into the portal vein and back to the liver [136], while the remaining BAs enter the colon, undergo biotransformation (deconjugation, dehydrogenation, and dehydroxylated) to the secondary BAs (deoxycholic acid, DCA and litocholic acid, LCA), and tertiary bile acid deoxycholic acid (DCA) by the resident gut microbiota. Secondary/tertiary BAs undergo passive diffusion and reabsorption to the liver through the portal vein [49]. In the ileocyte, BA uptake, intracellular transport and secretion into the portal vein require the apical sodium dependent bile acid transporter (ASBT), the cellular intestinal BA binding protein (I-BABP), and the basolateral heterodimeric organic solute transporter (OSTα/β), respectively. BAs are minimally (<5%) lost in the feces and this amount equals the amount synthetized daily and secreted in the liver. About 10–50% of re-absorbed BAs undergo peripheral spillover into systemic circulation [137]. The liver recycles absorbed BAs and secretes them back to the biliary tract. This is the so-called enterohepatic circulation which, several times every day, is the exchanging mechanism between the gut and the liver.

In the terminal ileum, primary BAs entering the enterocyte, act on the orphan farnesoid-X receptor (FXR = NR1H4) leading to increased enterokine fibroblast growth factor 19 (FGF19) that enters the portal circulation with effects on the gallbladder and liver. In the liver, FGF19 binds to FGF receptor 4 (FGFR4)/β-Klotho, a step activating c-Jun N-terminal kinase/extracellular signal-regulated kinase (JNK/ERK) signaling, which inhibits expression of CYP7A1 and CYP8B1 and hepatic BA synthesis, in synergy with the FXR-SHP inhibitory pathway [124,138]. BAs enter the liver via sodium taurocholate cotransporting polypeptide (NTCP) and organic anion transporting polypeptide (OATP), acting act as physiological nuclear ligands for FXR, which regulates target gene transcription by binding to RXRs as a heterodimer [139]. This results in increased transcription of the small heterodimer partner (SHP) expression. SHP, in turn, inhibits LRH-1, preventing the activation of target genes that participate in BA and fatty acid synthesis. In the absence of BAs, LRH-1 acts together with LXR to stimulate BA synthesis [140,141,142]. FXR also regulates the enzymatic activity that is involved in BA conjugation to glycine or taurine and hepatic BA secretion by of BSEP, and hepatic phospholipid secretion by ABCB4. BAs re-entering the liver also interact with the liver GPBAR-1 expressed in Kupffer cells, in concert with the pathway activated by the FGFR4/β-Klotho. FXR activation also coordinates BA detoxification enzymes (i.e., cytosolic sulfotransferase 2A1 (SULT2A1), aldol-keto reductase 1 B7 (AKR1B7), cytochrome P450 3A4/3a11 (CYP3A4/Cyp3a11), and UDP-glycosyltransferase 2B4 (UTG2B4)) [143].

In the ileum, BAs also activate the ileal G protein-coupled receptor (GPBAR-1 = TGR5), leading to secretion of three hormones, peptide YY (PYY), glucagon-like peptide 1 (GLP-1), and glucagon-like peptide 2 (GLP-2). This interaction has metabolic effects on glucose metabolism [144], insulin metabolism [145], energy expenditure [146], anti-inflammatory immune response [147], and appetite via GPBAR-1 receptors found in brown adipose tissue and muscle [144,148]. BAs are also excreted from the hepatocyte into the portal vein via specific transporters, i.e., the resistance protein 3 and 4 (MRP3, MRP4) and OSTα/β. From the peripheral circulation, BAs also undergo renal uptake by the apical sodium/dependent bile acid transporter (ASBT) in the proximal tubule. MRP 2, 3, 4 transporters regulate the glomerular filtration of BAs [149].

In the intestine, micellization of BAs contribute to the digestion and absorption of cholesterol, triglycerides, and fat-soluble vitamins. BAs also are signaling molecules in modulating epithelial cell proliferation, gene expression, and lipid and glucose metabolism. Thus, the enterohepatic circulation of BAs as signaling molecules produces profound effects on lipid and glucose metabolism.

### 3.4. Intestinal Monocellular Layer

A fourth level of the gut barrier is the monocellular layer [150], which include enterocytes, Goblet cells (which produce mucus), Tufts cells (with chemosensory function) [151], and Paneth cells (which produce antimicrobial peptides) [152]. This cellular barrier has physical, electrical, and chemical properties. The layer is impermeable to most solutes that need a specific transporter to cross the barrier, a mechanism involving transcellular pathway. Intercellular spaces are closed by the presence of specific apical junctional complex, i.e., a tight junctions (TJs) and the adherens junctions. Over 40 proteins contribute to a TJ (i.e., claudins, peripheral membrane proteins, zonula occludens (ZO) 1 and 2, occluding, E-cadherins, and desmosomes) [153,154,155]. The cytoskeleton connects TJs and adherens junctions, and TJs contribute to active and passive transport through the gut barrier [156]. TJs regulate the passive flow of the solutes and water through the paracellular pathway, and operate as a size- and charge-selective filter [157]. The two different routes involve the leak pathway (transport of larger substances namely proteins and bacterial components) and a second claudin-mediated pathway, which limits the flow to small (<4 Å) molecules. Due to the active transport across cells sealed with TJs, the human body prevents the uncontrolled translocation of substances and allows an active transcellular transport through the enterocytes [158]. Moreover, cytokines, tumor necrosis factor-alfa (TNFα), interferon gamma (IFNγ) signaling kinases, and cytoskeleton (myosin light chain kinases (MLCK)) contribute to regulate TJ [159,160,161,162]. Such mechanisms might be impaired during liver disease [163,164,165]. The negative charge of brush border microvilli (depending on polar carbohydrates and charged transmembrane proteins) opposes the negative charge of bacterial cell wall and inhibits bacterial translocation [166].

We assessed urine recovery of orally administered sucrose, lactulose/mannitol, and sucralose by using triple quadrupole mass-spectrometry and high-performance liquid chromatography. We found that increased colonic (but not stomach and small intestinal) permeability was linked to obesity and liver steatosis, regardless dietary habits, age, and physical activity [167] (Figure 6).

### 3.5. Immunological Gut Barrier

A fifth level of the gut barrier is the immunological barrier, starting from the secretory ability of the Paneth cells to secrete several antimicrobial peptides (i.e., defensins, cathelicidines, resistin-like molecules, bactericidal-permeability-inducing proteins and lectins, and IgA immunoglobulins) into the gut lumen at the inner face of the mucin layer hosts [93,168]. A further barrier is the population of intraepithelial cells (conventional and unconventional αβ and δχT cells and mononuclear phagocytes [169]) and lamina propria cells.

Intraepithelial lymphocytes express type I cytokines and act as first-line cytolytic defense. These lymphocytes release antimicrobial peptides in response to cytokines released by intestinal epithelial cells or by engaging in the activation of NK cell receptors expressed by intestinal intraepithelial lymphocytes lineages [169]. Mononuclear phagocytes have protrusions that act as direct sensors of the intestinal lumen [170,171] while developing oral tolerance after delivering food antigenic peptides to lamina propria dendritic cells [172].

Immune cells act as second line of defense and contribute to tissue regeneration when the lamina propria is damaged. The cells are highly specialized to recognize microbial antigens or metabolites. Cells consist of CD4+ T cells lymphocytes, innate-like cells (iNKT cells), and mucosa associated invariant T cells.

Lipids presented on CD1 molecules are recognized by NKT cells [173]. Mucosa-associated invariant T cells will interact with raboflavin metabolites presented by MR1 molecules [174,175]. CD4+ T cells include two populations of cells.Th17 cells release IL17-A, IL17-F, and IL-22 (which contributes to strengthen tight junction molecules and stimulate epithelial cell regeneration [176]); IgA production; and the expression of pIgR, which allows their translocation towards the gut lumen [177,178]. These cells accumulate if segmented filamentous bacteria reaches the intestinal epithelium [178,179,180].T regulatory cells are thymic derived cells (which recognize self-antigens and control the function of autoreactive T cells) and peripheral derived T regs (which recognize food antigens in the small intestine or microbial antigens in the large intestine), where they control tolerance to innocuous non self-antigens [181].Innate lymphoid cells, also found in the intestine, rapidly release type 1, 2, and 3 cytokines in a response to infection before the response of adaptive T cells [116,182].


There is a precise equilibrium between the gut and the resident microbiota because pattern-recognition receptors (Toll-like receptors, TLRs, and nucleotide binding oligomerization domain-like receptors) on the intestinal epithelial cells will recognize MAMPs and PAMPs crossing the intestinal barrier. In this scenario, recruited dendritic cells transport the given antigens to the mesenteric lymph nodes (MLNs) for antigen presentation. Following this step, priming and maturation of B and T lymphocytes occurs, as part of the adaptive immune response in the gut associated lymphoid tissue [183,184]. As mentioned earlier, the microbiome takes part in the maturation of several immune cells [116,117].

### 3.6. Gut Vascular Barrier

A sixth level of the gut barrier is the gut-vascular barrier [185] with similarities between blood-brain barrier and intestinal barrier [185,186]. This barrier should prevent the translocation of bacteria and/or microbial components across the extracellular and the intestinal epithelial barrier. The so-called gut-vascular unit includes the intestinal endothelium associated with pericytes and enteric glial cells, TJs, and adherens junctions (permeable to most of the small nutrients) [185,186,187]. Endothelial plasma membrane are isolated and carry active and passive transporters, while glial cells contribute to gut homeostasis intestinal permeability. The intact endothelium allows the free diffusion of 4 kD dextran, but 70 kD dextran is blocked. The gut-vascular barrier ultimately blocks a further dissemination of bacteria to liver and spleen, blocked by a second barrier [185]. Under certain circumstances (i.e., infection with *Salmonella enterica serovar Typhimurium*), the gut-vascular barrier becomes disrupted and larger substances can permeate, even in the absence of inflammation and vasodilation. This dissemination occurs through the portal circulation more than the lymphatic vessels. A marker of endothelial permeability (vesicle-associated protein-1 (PV1)) can increase during such events and bacteria can be found systemically [185]. The intestinal barrier is typically disrupted in conditions like celiac disease [185], anchylosing spondylitis [188], and NASH [87].

### 3.7. The Liver Barrier

In the health condition, the translocation of small amounts of microorganisms and bacterial products is an ongoing process towards mesenteric lymph nodes (MLNs) [168]. Here, the immune system is continuously stimulated, modulated to achieve immune tolerance [189,190], which allows the killing of microbes without significant inflammatory modifications at a systemic level [191,192]. Minimal amounts of bacterial mRNAs and lipopolysaccharide (LPS) [88,193,194] can permeate to the liver and help detoxifying bacterial products [195,196]. Nevertheless, the liver is usually free of bacteria [195], while acting as a second firewall for microorganisms penetrated after mucosal damage and escaped from MLNs surveillance activity [88,195,196]. At the level of the hepatic sinusoids Kupffer cells account for over the 80% of all tissue macrophages and contribute to phagocytize and kill microbes that originate from the bloodstream [88,195,197,198,199]. Liver resident macrophages also participate in the clearance of microorganisms and MAMPs and PAMPs. Kupffer cells are able to process *E. coli* endotoxin [200]. Kupffer cells depleted mice display delayed clearance of *E. coli* K-12 during bacteremia [195], while Kupffer cells perform phagocytosis and killing of green fluorescent protein expressing *B.burgdorferii* and antigen presentation to natural killer (NK) cells [197]. Mononuclear cells, via functional Toll-like receptor 4 (TLR4) [201,202], are activated by the lipopolysaccharide binding. The LPS-LBP complex stimulates liver resident myeloid cells via mCD14 and TLR4 [203,204,205].

## 4. The Main Constituents of the Ongoing Liver Damage

Several studies in animal models suggest that during liver damage, changes include composition of gut microbiota, quality/quantity of mucus, gastrointestinal motility, epithelial barrier, TJs, and intestinal immunity.

Disruption of the intestinal barrier increases the intestinal permeability, leading to filtration of damaging agents like MAMPs and PAMPs (LPS, microbial DNA, peptidoglycans, and lipopeptides), metabolic products, and whole bacteria to local MLNs, where clearance is defective [206,207,208,209]. The liver is then reached by these agents/bacteria via the mesenteric and portal circulation [88], and detrimental agents perpetuate the local damage in the liver but could also activate a systemic inflammatory response [210,211,212,213,214]. Activated Kupffer cells are essential in this respect [199,212,215,216,217].

### 4.1. Inflammation

Both PAMPs and TLRs interaction activate intracellular molecular pathways (MyD88-dependent or MyD88-independent) that, in turn, activate NF-κB and inflammatory cytokines including TNF-α, IL-1β, IL-6, IL-12, IL-18, chemokines like CXCL1, CXCL2, CCL2, CCL5, CCL3, CCL4, vasoactive factors like nitric oxide (NO), and production of reactive oxygen species (ROS) [218]. Liver inflammation is further enhanced following the recruitment of systemic leukocytes (CD4+ T cells, neutrophils, and monocytes) [212,215], with induction of apoptosis and necrosis of the hepatocyte [219], activation and proliferation of hepatic stellate cells (HSC), and production of transforming growth factor-β (TGFβ) with activation of fibrosis [217,220]. The following upregulation of the expression of matrix metalloproteinases (MMPs) enhance the destruction of the hepatic tissue [221,222] in tandem with increased expression of tissue inhibitors of matrix metalloproteinases (TIMPs) a step leading to inhibition of degradation of collagen fibrogenesis in the liver [221,222,223,224]. Indeed, TIMP-1 is a predictive marker for NASH [225].

### 4.2. Oxidative Stress

Oxidative stress contributes to liver damage [226], including liver steatosis [227], since hepatocytes are sensitive to oxidative stress-related molecules [226,228,229]. ROS can deteriorate the ongoing equilibrium of the gut-liver axis and be responsible for intestinal barrier damage. Several conditions can impair the intestinal permeability and act via abnormal redox state in the gut, including diet [230], alcohol [231], infectious [232]/primary inflammatory diseases [233], and drugs [234]. An ongoing hypoperfusion-dependent hypoxia of the intestinal mucosa leads to increased activity of xanthine oxidase, more ROS release and oxidative damage [235]. A vicious circle arises from activation of TLR of Kupffer cells and ROS generation [236], as well as activation and proliferation of hepatic stellate cells (HSC) [237]. In response to ROS, Kupffer cells produce cytokines and chemokines, and stimulate HSCs [229].

Protective mechanisms include the release of macrophagic IL-10 at the intestinal mucosa (modulation of the innate immune activation, decreased tissue damage, amelioration of integrity of the gut barrier, and decreased endotoxin absorption [238,239], and in the liver (reduced inflammations and fibrosis, and decreased activation of Kupffer cells functions [240,241]. NK cells also contribute by killing early activated and senescent HSCs, a step leading to limitation of fibrogenesis [242,243].

### 4.3. The Intestinal Microbiota

Gut microbiota, a super-organism hosted in the human body, can be involved in the pathogenesis of NAFLD [85]. Changes in microbiota signature might exist in NAFLD patients, but findings are difficult to be comprehensively elucidated in different NAFLD phenotypes. The difficulties in interpreting the results depend on the coexistence of complex interactions between genetic and environmental factors across several metabolic abnormalities, which also involves the liver.

For example, BMI is a determinant of gut microbiota [244]. Because NAFLD is often associated with overweight and obesity, the involvement of gut microbiota in the pathogenesis of NAFLD is a matter of active debate. Indeed, the liver has a complex vascular system. The majority of the blood flow reaches the liver from the intestine through the portal vein. Changes of intestinal immune system may affect intestinal permeability and bacterial translocation. This step might be responsible for a cascade of disorders including obesity, metabolic, and liver diseases [245,246]. Specific profiles of the gut microbiome might influence the inflammatory and fibrosis responses in NAFLD patients [247]. The combination of dysbiosis and increased intestinal permeability allows gut microorganisms to promote liver damage by release of MAMPs and PAMPs (i.e., LPS) or by products of their metabolism (i.e., ethanol, SCFAs, and trimethylamine) that enters the portal vein, which contributes with about 70% of the blood entering the liver [248,249,250]. Changes in the quality and/or quantity and/or topographic distribution of the microbiota might contribute to promote such abnormalities. This step promotes the activation of both Kupffer cells and hepatic stellate cells, for example following the influx of the LPS which acts on cell Toll-like receptor 4. Few animal studies point to this important pathway linking intestinal lumen, barrier, and liver damage. If the LPS is injected to rats, a picture of steatohepatitis develops. By contrast, anti-tumor necrosis factor antibodies improve the steatosis [251,252]. In addition, genetically obese mice develop increased intestinal permeability, a condition promoting increased portal endotoxemia [193,253]. The study of Imajo et al. [254] suggests that obesity-induced leptin plays a crucial role in NASH progression via enhanced responsivity to endotoxin (and therefore bacteria-derived products). Upregulation of CD14 in Kupffer cells and hyperreactivity against low-dose LPS occurred in high-fat diet (HFD)-induced steatosis mice, but not chow-fed-control mice. Accelerated NASH progression (with liver inflammation and fibrosis) occurred in this condition. In chow-fed mice Leptin increased hepatic expression of CD14 via STAT3 signaling and hyperreactivity against low-dose LPS without steatosis, while leptin-deficient ob/ob mice with severe steatosis had a marked decrease in hepatic CD14. In different mouse models, Henao-Meja et al. [255] investigated the role of innate immunity inflammasomes NLRP6 and NLRP3 and the protein IL-18 on NAFLD/NASH progression, with respect to the gut microbiome. Changes associated with inflammasome-deficiency influenced the configuration of the gut microbioma and were associated with worsening of hepatic steatosis and inflammation. Findings parallel the influx of TLR4 and TLR9 agonists into the portal circulation. This influx, in turn leads to enhanced hepatic tumor-necrosis factor (TNF)-alpha expression and NASH progression. Within a scenario of defective NLRP3 and NLRP6 inflammasome sensing, the microbiome appears to play a role connecting systemic auto-inflammatory and metabolic disorders. There is an association between NAFLD, SIBO, and increased endotoxemia [165,256,257,258]. The ongoing dysbiosis might encompass a condition named SIBO [259,260]. The occurrence of SIBO was higher in NAFLD [165]/NASH [257] patients than healthy controls.

NAFLD patients with advanced fibrosis show increased abundance of *Escherichia coli* and *Bacteriodes vulgatus* [261]. Obese children with NASH had greater abundance of the genus *Escherichia* [262]. Changes of intestinal microbiota in NAFLD may also occur because of dietary factors. A Western diet, enriched in fat, proteins of animal origin and simple sugars promotes *Bacteroides* abundance. By contrast, a diet rich in fiber and indigestible plant polysaccharides increases *Prevotella* abundance. Notably, *Bacteroides* genus correlates with NASH while *Prevotella* abundance decreased in NASH [247,263]. *Ruminococcus* genus abundance increases with significant liver fibrosis (score ≥F2) in humans [247]. In animal models this genus correlates with the development of metabolic impairment [264].

In obesity the ratio *Firmicutes*:*Bacteroidetes* increases, compared to lean subjects on the same diet [255]. Shift of microbiome in blood is reported in patients with liver fibrosis [265], as well as in metabolic diseases [266,267]

Yun et al. [26] had the idea of investigating the association between fecal and blood microbiome profiles, and the presence of NAFLD in obese versus lean Korean subjects. They sequenced the V3-V4 domains of the 16S rRNA genes. The 176 NAFLD subjects had distinct bacterial community with a lower biodiversity and a far distant phylotype, compared with 192 controls. In the gut, the microbiome showed decreased *Desulfovibrionaceae* in lean NAFLD but not in obese NAFLD. In the blood, *Succinivibrionaceae* showed opposite correlations in the lean vs. obese NAFLD. In addition, both gut and blood *Leuconostocaceae* was associated with the obese NAFLD. This study suggests that fecal and blood microbiome profiles showed different patterns between subjects with obese and lean NAFLD. Such profiles might therefore be potential biomarkers to discriminate diverse NAFLD phenotypes. In NAFLD, Raman et al. described decreased members of *Firmicutes* [268]. A study looked at patients with biopsy-proven NAFLD (*n* = 57) and reported a twofold increase in NASH in patients with *Bacteroides* genus counts in the 2nd and 3rd tertile, as compared with patients with lower *Bacteroides* counts. This latter group of patients had an abundance of *Prevotella* bacteria. Moreover, fibrosis could be affected by changes in microbiota, since patients with *Ruminococcus* counts in the third tertile had a twofold increase in stage 2 or greater fibrosis, as compared with patients with lower levels of *Ruminococcus* [261,269]. In another study, NAFLD patients had a lower percentage of species *Bacteroidetes* and higher levels of *Porphyromonas* and *Prevotella* than healthy subjects. Notably, the predisposition to NAFLD related to expression of Toll-like receptors (TLR) 4 or 9, or the receptor of tumor necrosis factor-alpha [262].

As mentioned earlier, the intestinal microbiota is essential during the process of enterohepatic circulation of primary BAs. At the level of the colon, the bile acids undergo complex biotransformation by the resident microbiota and are transformed into secondary BAs [12,89,270]. Under specific circumstances, the dysbiosis may contribute to hepatocellular injury because of increased deconjugation of BAs (i.e., production of more cytotoxic secondary bile acids) and inactivation of hepatic lipotropes (i.e., choline). Indeed, a choline-deficient diet is one way to develop NAFLD in the rat model [227,271,272,273,274].

At the extreme of the NAFLD spectrum is the so-called “cryptogenic” cirrhosis. Of note, low gut microbiota diversity exists in NAFLD-related liver cirrhosis compared to healthy controls. At the genus level, *Bacteroides*, *Ruminococcus*, *Klebsiella*, *Prevotella*, *Enterococcus*, *Hemophilus*, *Lactobacillus*, *Streptococcus*, *Pseudomonas*, *Phascolarctobacterium*, *Veillonella*, *Atopobium*, *Parabacteroides*, *Dialister*, and *Christensenella* were more abundant, while *Methanobrevibacter* and *Akkermansia* decreased [275]. Changes, however, might be influenced by the advanced status of disease, portal hypertension, drugs, etc.

### 4.4. Altered Intestinal Permeability

Changes to the aforementioned conditions might originate following impaired intestinal permeability due to disrupted of tight junctions, dysbiosis, and/or small intestinal bacterial overgrowth (SIBO). These conditions are associated with the spectrum of liver disease, i.e., alcoholic and non-alcoholic fatty liver [165,276]. Children with NAFLD showed a positive correlation between altered intestinal permeability and liver inflammation/fibrosis [256]. A metanalysis shows that patients with NAFLD more frequently exhibit altered intestinal permeability [277].

Small intestinal bacterial overgrowth (SIBO) can develop in NAFLD patients [165,278,279,280,281] along with dysbiosis [247]. Abnormal intestinal permeability might therefore be involved in this scenario [45] and act as a factor leading to liver inflammation and fibrosis. Several animal and human studies evidences point to a role for intestinal permeability in the pathogenesis of NAFLD.

In the steatotic mice endotoxin triggers liver inflammation [282]. Obese mice (C57BL/6Job/ob genetically leptin deficient and C57BL/6Jdb/db functionally deficient for the long-form leptin receptor) show increased epithelial permeability to horseradish peroxidase. TJ proteins (ZO-1 and Occludin) distribution was abnormal in both types of obese mice that also exhibited increased circulating levels of endotoxin in portal circulation and levels of circulating proinflammatory cytokines (IL-1, IL-6, INF-γ, and TNF-α) than controls. HSC became activated and displayed enhanced sensitivity to LPS and released higher levels of cytokines [193]. F11r−/− mice fed a steatogenic diet (high saturated fat, fructose and cholesterol for 8 weeks) developed a form of severe steatohepatitis with liver inflammation (hepatocyte ballooning and inflammatory cells infiltration), fibrogenesis, and increased in serum transaminases, as compared with control animals [250]. Notably, the murine gene *F11r* encodes the junctional adhesion molecule A (JAM-A), which is a constituent of the TJs and which modulates the epithelial barrier function, regulating IP and inflammation [283,284,285,286].

When inducing gut inflammation and gut-vascular barrier dysfunction with dextran sulfate sodium (DSS) in C57BL/6 mice fed a high-fat diet (HFD) for 12 weeks, liver fat vacuoles and leukocyte infiltration were more represented in DSS and HFD-fed mice compared to HFD-fed mice. Several other alterations included increased levels of hepatic mRNA coding for inflammatory cytokines (IL-1, IL-6, TNF-α, MCP-1), higher expression of collagen I and profibrogenic factors mRNA (TGF-β, Actin α2, tissue inhibitor of metalloproteinase-1, and plasminogen activator inhibitor-1). Additional modifications include upregulation of TLR4 and TLR 9, downregulation of ZO-1 and Claudin-1 and increased expression of PV1 [287]. Human studies are also instructive in this field and point to a role for increased intestinal permeability, disrupted TJs, and SIBO in NAFLD patients. A previous trial compared 22 patients with biopsy-proven NASH and 23 controls subjects. The authors assessed overgrowth by combined (14)C-D-xylose and lactulose breath test, intestinal permeability by a dual lactulose-rhamnose sugar test, serum endotoxin levels by limulus amoebocyte lysate assay and TNF-alpha levels by ELISA. Results showed that 50% of patients and 22% of controls (*p* = 0.048) had SIBO. Whereas intestinal permeability and serum endotoxin levels were similar in the two groups, endotoxin assay confirmed significantly higher TNF-α levels in patients than controls (14.2 and 7.5 pg/mL, respectively, *p* = 0.001) [257].

Another study compared patients with biopsy-proven NAFLD (*n* = 35) patients with untreated celiac disease (*n* = 27, as a model of intestinal hyper-permeability), and healthy subjects (*n* = 24). The authors used glucose hydrogen breath testing to diagnose SIBO and urinary excretion of (51)Cr-ethylene diamine tetraacetate ((51)Cr-EDTA) test to diagnose intestinal permeability. Orally administered (51)Cr-EDTA is not metabolized and is poorly absorbed (1–3%) from the gastrointestinal tract. In the presence of TJs disruption and (51)Cr-EDTA crosses the intestinal barrier through the paracellular pathway [123,288,289]. The authors also used immunohistochemical analysis of zona occludens-1 (ZO-1) expression in duodenal biopsy specimens to diagnose the integrity of TJs within the gut. NAFLD patients had significantly increased intestinal permeability and SIBO (3x) compared to the control [165]. (51)Cr-EDTA excretion levels and SIBO prevalence increased with the degree of liver steatosis. In addition, by duodenal histology NAFLD patients had reduced ZO-1 expression. Increased intestinal permeability and SIBO were independent with the severity of liver inflammation, fibrosis, and NASH.

NALFD children had increased ratio between urinary excretion of orally-administered lactulose and mannitol (L/M ratio) as a marker of intestinal permeability [256,257,290]. L/M ratio further increased in NASH patients. The increased LPS pointed to bacterial translocation while the extent of hepatic inflammation and fibrosis was proportional to the degree of intestinal permeability [256].

### 4.5. Products of Microbiota

Microbe-associated molecular patterns: the intestinal microbiome is an invaluable source of gases, metabolites, and bacterial products, namely MAMPs and PAMPs. Some of such metabolites deriving from the interaction of the microbiota with endogenous and exogenous substances might act as effectors of damage in the liver after crossing the portal circulation. Upon uptake in the liver, MAMPs activate local inflammatory changes mediated by patter-recognition receptors (PRRs) located on the hepatic stellate cells [291] and Kupffer cells [217]. In turn, endotoxins will activate TLR4, TLR9 (by methylated DNA) and TLR2 (by Gram-positive bacteria) [162], which represent the first step of immune response in liver disease. Further steps include the activation of further inflammatory events via activation of nuclear factor- χB (NF-χB) by the myeloid differentiation primary response protein (MYD88) enhanced hepatic tumor-necrosis factor (TNF)-alpha expression that drives NASH progression [255]. In hepatic stellate cells, fibrosis is promoted via TLR4 signaling that downregulates BMP and activin membrane-bound inhibitor homologue (BAMBI, a decoy receptor for transforming growth factor-β (TGFβ)) [218]. Further steps include the release of inflammatory cytokines, oxidative stress, and endoplasmic reticulum stress, all factors initiating and perpetuating liver damage [292]. Evidences in animals show that diets enriched in fat or deficient in choline drive steatogenic, inflammatory, and fibrogenic hits requiring TLR-4 or 9 [293,294,295]. In addition, inflammasome-deficiency-associated changes in the configuration of the gut microbiota parallel exacerbated hepatic steatosis and inflammation.

Alcohol: Dietary ethanol crosses the gastrointestinal mucosa by simple diffusion in the stomach and small intestine (about 20% and 70%, respectively) [296]. The liver is involved in ethanol metabolism pathway with enzymes that transform ethanol to acetaldehyde (enzyme alcohol dehydrogenase) and to acetate (enzyme acetaldehyde dehydrogenase) [297,298]. Most of the alcohol in the intestine is from the systemic circulation, while a smaller part originates as a product of luminal microbial fermentation, as both bacteria and enterocytes are equipped with alcohol-metabolizing enzymes [298,299]. A study showed an increase in hepatic expression of ethanol-metabolizing genes in germ-free mice and exacerbation in hepatic steatosis [300]. Acetaldehyde impairs the intestinal TJs and this, in turn, compromises the gut barrier enabling translocation of microbial products [301,302,303]. Additional effects include downregulation of the expression of intestinal AMPs [304,305] and activation of a host inflammatory immune responses [306,307,308]. Together with reduced intestinal levels of butyrate [309,310,311], changes increase intestinal permeability [312,313,314,315]. Humans display high concentrations of endogenous alcohol production [316]. Similar findings occur in animals with intestinal blind-loops [317]. Obese females with *Candida albicans* overgrowth have increased breath alcohol levels after a carbohydrate load [318,319]. Notably, colonic bacteria and yeast are able to produce both ethanol and acetaldehyde [320]. Even if ethanol concentrations remain low, the microbiota can oxidize ethanol to high concentrations of acetaldehyde, which is promptly absorbed into the portal blood stream. These pathways initiate histological changes similar to those occurring in NAFLD [321]. *Proteobacteria*, particularly *Enterobacteriaceae*, ferment carbohydrates to ethanol [262], and the amount produced can be remarkable [322]. A correlation exists between abundance of *Proteobacteria, Enterobacteriaceae, Escherichia,* liver inflammation (NASH), and serum levels of alcohol in children [262]. In another complementary study, fasting ethanol levels were positively associated with insulin resistance and were significantly higher in children with NAFLD than in controls. The authors suggest that increased blood ethanol levels in patients with NAFLD may result from insulin-dependent impairments of ADH activity in liver tissue rather than from an increased endogenous ethanol synthesis [323]. *Ruminococcus* ferments complex carbohydrates with production of ethanol and this property may be responsible for further liver damage [324]. Gut-derived endogenous alcohol might therefore participate in mechanism of liver damage. Increased luminal and circulating levels of ethanol, acetaldehyde, and acetate occur in non-alcoholic and alcoholic liver disease [165,325], and metabolites are independently associated with liver damage [326,327,328]. Increased ethanol production activates hepatic ethanol metabolic pathways and oxidative stress [329], contributing to the pathogenesis of NASH [262].

Bile acids: As inferred from the previous description, the gut microbiome can shape the final profile of BAs. Intestinal bacteria are responsible for the transformation of the primary BAs to the secondary BAs by deconjugation, oxidation of hydroxyl groups in 3, 7, and 12 positions, and 7-dehydroxylation [330]. A bidirectional cross-talk exists between BAs and the intestinal microbiome [331]. BAs (mainly the secondary DCA) display antimicrobial properties and can modulate the species of intestinal microbiome. The detergent effects of DCA acts on bacterial cell membranes and influences the integrity of the bacteria while modulating microbial populations [332]. In addition, BAs binding to FXR promote the production of peptides with antimicrobial effect (AMPs) (e.g., angiogenin 1, RNAase family member 4 [333,334]), which prevent gut barrier disruption and dysfunction. This process has an influence on the hydrophobicity of the BA pool and ongoing dysbiosis initiates a series of events due to unbalance between primary/secondary BAs ratio. Ultimately, changes in hepatic bile synthesis and metabolic abnormalities can arise, including FXR-dependent effect on intestinal barrier and inflammation [335], metabolic pathways [336], and carcinogenic effects [337]. Notably, FXR-deficient mice do not become responsive to diet-induced obesity [336], likely due to mechanisms related to both microbiome [334] and intestinal FXR [338]. Evidences support this complex balance between BAs and microbiota. The obstruction of bile flow is a predisposing factor for bacterial overgrowth and translocations of bacteria within the intestine. Oral administration of BAs in the mouse model can reverse this condition. The FXR-induced gene expression might modulate this pathway associated with the enteral protection and the inhibition of bacteria damage to the intestinal mucosa [333].

The abnormal intestinal microbiota is parallel to the altered bile acid homeostasis, which contributes to the metabolic dysregulation seen in NAFLD. This step can lead to impaired BA signaling in the intestine and the liver, as seen in mice on a high-fiber diet and in NAFLD patients. The increased synthesis of BAs become evident, as increased serum concentrations of primary and secondary BAs [339,340], increased hepatic bile acid synthesis, and total fecal bile acids and primary to secondary bile acid ratio [341]. Studies suggest that the microbiota, while governing the production of the secondary BAs, also influences FXR-mediated signaling in the intestine and the liver. Abundance of bacteria producing the secondary DCA might lead to a DCA-dependent suppression of FXR- and FGFR4-mediated signaling [11,339]. Notably, intestinal FXR expression decreases in mice on a high-fiber diet. Obeticholic acid restores the integrity of the gut vascular barrier and reduces the portal influx of PAMP to the liver [87].

Fatty acids: in health, the colon microbiota produces short chain fatty acids (SCFAs) acetic, propionic, and butyric acid as the main SCFAs following fermentation of indigestible carbohydrates, i.e., dietary fiber [342]. SCFAs provide the energetic substrate for colonocytes, contribute to the maintenance of the gut barrier integrity, while controlling intestinal inflammation and satiety [343,344]. SCFAs are absorbed in the intestine through the portal circulation and, in the liver, provide the necessary energy source used in gluconeogenesis and lipogenesis [248,345,346]. Moreover, at the intestinal enteroendocrine L cells, SCFAs interact with G protein coupled receptors GPR41 and GPR43 and stimulate the release of the peptide YY (PYY). This hormone slows gastric emptying and intestinal transit and helps energy absorption [347]. In addition, the release of glucagon-like peptide-1 enhances glucose-dependent insulin release [348]. Patients with type 2 diabetes mellitus exhibit reduced abundance of butyrate-producing bacteria [349], while SCFAs supplementation in patients with type 2 diabetes mellitus increased the abundance of butyrate-producing bacteria. This effects was associated with increased GLP-1 and hemoglobin A1c levels [350]. Increased levels of acetate might represent a marker of increased production of endogenous ethanol in the intestinal lumen and liver metabolism (see below). Decreased levels of butyrate might be a marker of tight junctions abnormality and increased intestinal permeability [309,312]. SCFAs appear protective in the gut. Mice fed with alcohol-enriched diet short-term and given tributyrin (the glycerol ester providing butyrate) improved the intestinal permeability and liver injury induced by ethanol [312].

Fasting-induced adipocyte factor (FIAF): The intestinal microbiota also inhibits the production and secretion of FIAF by the intestinal L cells and enterocytes. FIAF inhibits lipoprotein lipase (LPL), activates LPS, and increases triglyceride accumulation in the liver and the adipocytes [351]. Increased hepatic lipid storage activates the carbohydrate responsive element-binding protein and the sterol regulatory element-binding protein-1, perpetuating fat accumulation [352].

Choline: The macronutrient choline is processed into phosphatidylcholine (lecithin) and plays a role in the assembly and excretion of very-low density lipoprotein (VLDL) from the liver. This step prevents the formation of liver steatosis due to triglycerides [353]. Choline is also involved in the development of brain, nerve function, muscle function, and metabolic pathways [354]. Choline deficiency is associated with decreased production and release of VLDL, and therefore triglyceride accumulation in the liver [355]. In the murine model, a choline-deficient diet leads to liver steatosis and oxidative stress [227,271,273,274,356]. Intestinal bacteria can use choline for the synthesis of trimethylamine (TMA). Bacteria of the taxa *Erysipelotrichia* metabolize choline to TMA and this step decreases the bioavailability of choline and increases the portal influx of TMA, its conversion to trimethylamine N-oxide (TMAO), with steatogenic effects [357,358,359,360]. Suppression of intestinal microbiota in atherosclerosis-prone mice inhibited dietary-choline-enhanced atherosclerosis [355]. In addition, NAFLD patients have increased intestinal metabolism of choline, choline deficiency, and abundance of *Erysipelotrichia* taxa [358].

Other products: 3-(4-hydroxyphenyl) lactate correlated with specific bacterial species (*Firmicutes*, *Bacteroidetes*, and *Proteobacteria*), as well as hepatic fibrosis [269]. Microbiome-derived products of branched-chain and aromatic amino acid metabolism can also play a role in NAFLD. Phenylacetic acid and 3-(4-hydroxyphenyl) lactate, both related to insulin resistance. Obese, non-diabetic patients with hepatic steatosis and inflammation had low microbial gene richness, and increased microbial genetic potential for processing dietary lipids and endotoxin biosynthesis from *Proteobacteria*. In addition, both aromatic and branched-chain amino acid metabolism was dysregulated [361].

## 5. The Microbiota as a Target of Environmental Factors Involved in NAFLD

The microbiome is the first interface between the environment and almost all the metabolic, biochemical, endocrine, and signaling pathways influencing the onset and progression of NAFLD.

The gut microbiota is strongly influenced by dietary habits [362], smoking [363], ethanol intake [364], drugs (i.e., antibiotics [255,349,365,366,367], liraglutide [368], metformin [369], curcumin [370], and environmental pollutants (i.e., heavy metals, persistent organic pollutants, volatile organic compounds, pesticides [62,371,372,373,374]).

These external agents act on the microbiota affecting the ability to maintain the integrity of the mucosal barrier [106], the healthy function of epithelial intestinal cells, the efficiency of the immune system [375], and the production of local peptides and immunoglobulins with antimicrobial function [96,376,377]. Thus, abnormalities in the gut–liver axis with the microbiome as the key player leads to abnormal regulation of intestinal and metabolic homeostasis, influencing the pathogenesis of NAFLD [85]. Conversely, protective effects and a possible reversal of liver fat accumulation and lipotoxicity could be driven by favorable environmental factors modulating composition and relative abundance of specific phylotypes.

### 5.1. Dietary Habits and Microbiota

Dietary habits can influence the microbiota, the intestinal permeability, the vascular barrier, the enhanced influx of endotoxins in the portal vein, low-grade liver inflammation, and ultimately NAFLD/NASH, insulin resistance, and metabolic disorders [378,379]. Changes are likely due to the initial modification of the microbiota, rather than the dietary pattern per se [87]. This topic, however, is still under discussion. A Western-style diet is rich in saturated fat and sugars, and might promote injury. In mice, a high-fat diet or fiber-deprived diets change the composition of intestinal microbiota and can damage the intestinal barrier through increased intestinal permeability, reduced thickness of the mucous layer, abnormalities of tight junction proteins of the epithelial barrier and low-grade intestinal inflammation [114,380,381].

In the NAFLD animal models following high-fat diet or high-fructose intake gut permeability increased [246,382].

Both saturated fat and fructose promote the pro-inflammatory microbiome and decreases SCFAs production, which is essential for intestinal barrier function. Increased recruitment of macrophages with cytokines. Thus, TNF-α can be associated with intestinal mucosal inflammation [383,384]. Decreased expression of TJ proteins and a higher permeability of the gut barrier is another potential step [385]. In the animal model, a high-fat diet or a high-fructose diet induces endotoxemia by altering intestinal ZO-1 and occludin [193,386,387,388]. If LPS levels increase in serum due to diet, a further step can be the metabolic endotoxemia with activation of TLR-mediated low-grade liver inflammation. This is a condition potentially associated with NAFLD and NASH [379]. Of note, dysbiosis will generate an altered microbiota able to cross the intestinal epithelium at the disrupted site and to alter the TJs proteins [87]. The resulting inflammatory events lead to reduced lamina propria Treg cells, increased production of IFN-γ (by Th1 and CD8+ T cells), and increased production of IL-17 (by γδ-T cells) [381]. In an integrated study involving animals and humans, the authors reported that mice with defects in intestinal permeability develop more severe steatohepatitis after a high fructose, cholesterol diet than control mice, and colon tissues from patients with NAFLD have lower levels of JAM-A (junctional adhesion molecule), and higher levels of inflammation than subjects without NAFLD [250]. In accordance with these findings, another study found that mice genetically deficient in Jam1 on a high-fat and -fructose diet had increased intestinal permeability, endotoxemia, and hepatic inflammation [389].

In NAFLD adolescents, fructose- but not glucose-enriched drinks together with meals during 24-h increased postprandial endotoxin levels [390].

A less pro-inflammatory diet could modulate also the intestinal permeability in NAFLD patients. The Mediterranean diet is enriched in fibers, mono- and polyunsaturated fatty acids, antioxidants, polyphenols, and phytochemicals. Following this diet, SCFAs-producing bacteria might increase in the intestine due to the diet-induced prebiotic effects [391]. A small group of patients with NAFLD and increased baseline intestinal permeability ((51)Cr-EDTA) underwent a trial consisting of 16 weeks of a Mediterranean diet and 16 weeks of a low-fat diet. With both diets, intestinal permeability did not improve [392]. More studies are clearly required in this field.

### 5.2. Antibiotics, Probiotics, and Other Pharmacological Agents

Under certain circumstances due to disease or use of antibiotics, the function of the microbiome can be disrupted and lead to energy storage and metabolic disorders, such as obesity, diabetes, and metabolic syndrome [255,349,365,366,367]. Treatment with antibiotics leads to qualitative/quantitative modifications of the intestinal microbiome, and therefore their potential role for liver damage. For example, administration of polymyxin B was associated with improved steatosis grades in both rats and humans on total parenteral nutrition. A similar protection was documented in alcohol-exposed rats [393,394,395]. Following the intestinal bypass surgery and the associated hepatic steatosis, metronidazole administration improved the hepatic picture [396]. In children the administration of some probiotic species caused increased levels of glucagon-like peptide-1 (GLP-1) and improvement in fatty liver [397]. Administration of probiotics during four weeks to NAFLD mice improved the steatosis, hepatomegaly, and nuclear factor kappa-beta activity [398].

In an animal model (ob/ob mice fed a high fat diet), liraglutide, a drug useful in the treatment of insulin-resistance, decreased the fat content in the liver, reversed steatosis, and modified the overall composition, as well as the relative abundance of gut microbiota phylotypes involved in the pathogenesis of NAFLD (i.e., reduced Proteobacteria and increased *Akkermansia muciniphila*) [368].

In mice, curcumin was also able to alter the composition of a number of operational taxonomic units previously correlated with hepatic steatosis, finally resulting in an attenuation of liver fat deposition and improving the integrity of the intestinal barrier [370].

In a different animal model (C57Bl/6J mice fed fat-, fructose-, and cholesterol-rich diet), protective effects were also noticed following metformin administration in terms of changed composition of gut microbiota and integrity of the intestinal barrier [369].

### 5.3. Food Contamination with Chemicals of Environmental Origin, Microbiota, and NAFLD

Several studies underscore the relationships between chemicals (in particular endocrine disrupting chemicals, persistent organic pollutants, heavy metals) introduced with contaminated food or through contaminated consumer products, and alterations in gut microbiota, metabolic pathways, and hepatic fat accumulation.

In humans, urine levels of Bisphenol-A (BPA), a widely diffused plasticizer with well-known endocrine disrupting effects, have been linked with NAFLD [77], independently from the presence of type 2 diabetes [399]. In animals, dietary exposure to BPA increases lipid content and fat accumulation in the liver, and simultaneously affects the gut microbiota (i.e., increased abundance of *Proteonacteria*, decreased abundance of *Akkermansia*). Exposed animals also show increased intestinal permeability, upregulated expression of Toll-like receptor 4, increased phosphorylation of nuclear factor-kappa B in the liver and increased liver inflammation [400]. The negative effect of endocrine disruptors is also possible during early life exposure. Male offspring from mice exposed to low doses of BDE-47, a polybrominated diphenyl ether (PBDE), during pregnancy and until postnatal day 21 (through lactation) showed a worsening of high fat diet-induced obesity and liver steatosis, metabolic disturbances and alterations of gut microbiota, shifting the microbial community from *Bacteroidetes* (decrease) towards *Firmicutes* (increase) [401].

Evidences from animal and epidemiologic studies link NAFLD with persistent organic pollutants (POPs) [402]. A cross-sectional cohort study in adults without viral hepatitis, hemochromatosis, or alcoholic liver disease (NHANES), correlated ALT elevation, a proxy marker of NAFLD, with blood levels of polychlorinated biphenyls (PCBs), lead, and mercury [403]. In adult female C57BL/6 mice, the oral exposure during 6 weeks to low-dose of the polychlorinated biphenyl 126 (PCB126), a typical persistent organic pollutant, affected gut microbiota (decreased phylum levels of *Firmicutes*, increased *Bacteroidetes*), and promoted dyslipidemia and NAFLD. Notably, several specific bacterial taxa positively and significantly related to metabolic alterations [404]. Another study reported a significant alteration of the *Firmicutes* to *Bacteroidetes* ratio following PCB126 exposure and found an upregulation of Cyp1a1 gene expression (a marker of aryl hydrocarbon receptor activation), as well as increased markers of systemic and intestinal inflammation. Findings depict the intestine as a target of PCB126 toxicity [405].

In mice, the enrichment of a standard chow diet with a mixture of six commonly used pesticides (i.e., boscalid, captan, chlorpyrifos, thiofanate, thiacloprid, and ziram) at doses corresponding to the tolerable daily intake (TDI) of each pesticide promoted for 52 weeks had significant metabolic alterations. Rodents increased body weight and adiposity, developed glucose intolerance and liver steatosis, and had perturbations of gut microbiota in terms of altered urinary concentration of microbiota–related metabolites [406]. In humans, a large survey from the NHANES III study (12,732 adults) linked urinary cadmium (Cd) concentrations secondary to environmental exposure with hepatic necroinflammation, NAFLD, NASH in men, and hepatic necroinflammation in women [407]. In mice, sub-chronic (10 weeks) exposure to low doses of Cd in drinking water increased hepatic triacylglycerol, serum free fatty acid, and triglyceride levels through altered gene expression. Changes in energy metabolism were paralleled by changes (both structure and abundance) in gut microbiome, with decreased *Firmicutes* and γ-proteobacteria, increased serum levels of LPS and liver inflammation [408]. Another study in mice showed that low dose Cd exposure in early life promotes an increased expression of hepatic genes involved in fatty acid and lipid metabolic processes, leading to life-long metabolic consequences. In male mice, these metabolic alterations developed together with altered microbiota composition after 8 weeks (increased *Bacteroidetes*, reduced *Firmicutes*) [409].

## 6. Conclusions

The molecular, anatomical, and functional link between gut functions, intestinal microbiota, and liver constitutes the gut–liver axis. This axis is exposed to environmental factors, with the microbiome being the main interface. Complex interactions involve bacteria and intestinal barrier, which comes in close contact with dietary factors, food contaminants, drugs, chemicals, and bile. These elements can modulate the pathways linking the gut with the liver, leading to both local and systemic effects. The interplay is bidirectional, since BAs produced in the liver regulate microbiome composition and intestinal barrier. By contrast, intestinal products regulate BA synthesis, hepatic glucose, and lipid metabolism. Local, short-, and long-term changes of this fine interplay due to external or intrinsic factors (e.g., altered intestinal microbiome, increased gut permeability, changes in luminal levels of BAs) at the gut–liver axis disrupts the ongoing homeostasis, paving the way to local and systemic inflammation, and to several liver diseases, including NAFLD.

Changes in the liver might also affect the innate immune cells with exposure to gut derived bacterial products (endotoxin and metabolites like ethanol and trimethylamine). This step is invariably associated with liver inflammation.

According to the available evidence, the majority of environmental factors leading to abnormalities in the gut–liver axis and, in turn, to liver and metabolic dysfunction, could be managed. Future studies should therefore target the modulation of gut-liver homeostasis with both preventive and therapeutic goals.

## Figures and Tables

**Figure 1 jcm-09-02648-f001:**
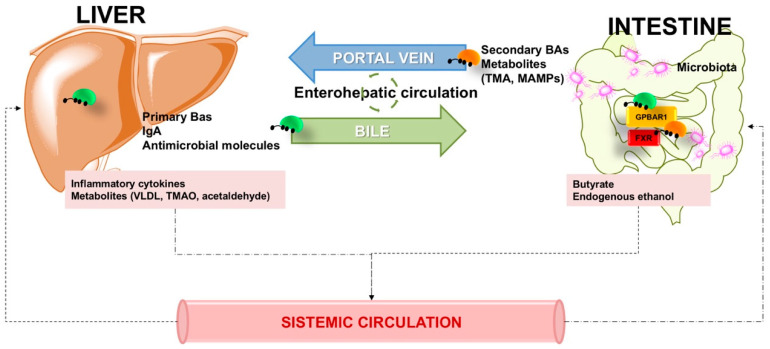
The communication between the liver and the gut is bidirectional. The liver secretes primary bile acids and antimicrobial molecules (immunoglobulin A (IgA) and angiogenin) into the biliary tract. Molecules reach the intestinal lumen and contribute to maintenance of gut eubiosis (while inhibiting intestinal bacterial overgrowth). During the enterohepatic circulation of bile, BAs act as signalling molecules by interacting with the nuclear receptor farnesoid X receptor (FXR) and the G protein-coupled bile acid receptor (GPBAR1). This interaction contributes to modulate hepatic bile acid synthesis, glucose metabolism, lipid metabolism and dietary energy utilization. Gut products include host and/or microbial metabolites and microbial-associated molecular patterns (MAMPs). The systemic circulation also plays a role, since products are translocated to the liver via the portal vein and can influence liver functions. The systemic circulation extends the gut–liver axis since liver metabolites from dietary, endogenous or xenobiotic substances (free fatty acids, choline metabolites and ethanol metabolites) are transported to the intestine through the capillary system. This mechanism of continuous recirculation of molecules across blood capillaries can distinctively affect the intestinal barrier. Butyrate, the short chain fatty acids, is protective (i.e. improves the colonic defensive border), while acetaldehyde can activate a barrier damage. Abbreviations: BAs, bile acids; MAMPs, microbial-associated molecular patters; TMA, trimethylamine; TMAO, trimethylamine *N*-oxide; VLDL, very-low density lipoprotein. Adapted from Tripathi et al. [11] and Di Ciaula et al. [12].

**Figure 2 jcm-09-02648-f002:**
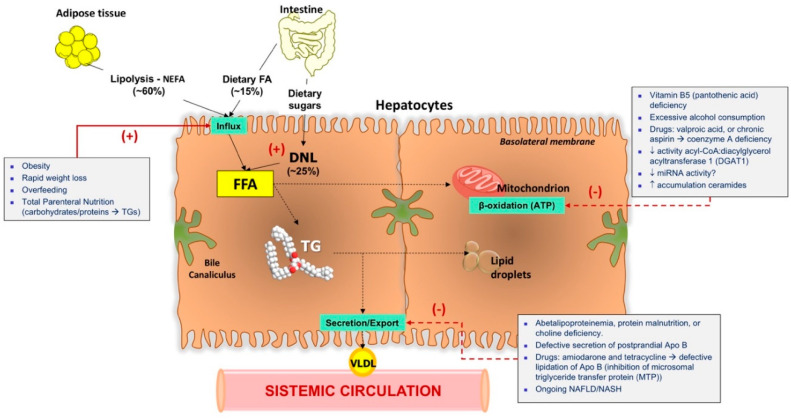
Mechanisms of lipid accumulation in the liver. In health, free fatty acids (FFA) originate from lipolysis of triglycerides (TG) in adipose tissue leading to circulating non-esterified fatty acids, (NEFA, ~60%), from dietary fatty acids (~15%), and from dietary sugars undergoing de novo lipogenesis (DNL, ~25%) mediated by several proteins (sterol regulatory element-binding protein 1, stearoyl CoA-desaturase, and fatty acid synthase). FA in hepatocytes undergo mitochondrial β-oxidation or re-esterification with glycerol to form TG. TG can be stored in lipid droplets or exported to blood as very-low density lipoproteins (VLDL). NAFLD is the consequence of excess flow of FFA and accumulation of intrahepatic TG due to several abnormalities depicted in the boxes, and associated with increased influx of FFA, increased DNL, decreased β-oxidation, and decreased secretion/export of TG as very-low density lipoproteins (VLDL). Abbreviations: DNL, de novo lipogenesis; FFA, free fatty acids; NEFA, non-esterified fatty acids; TG, triglycerides; VLDL, very-low density lipoproteins. (+), increased; (−), decreased.

**Figure 3 jcm-09-02648-f003:**
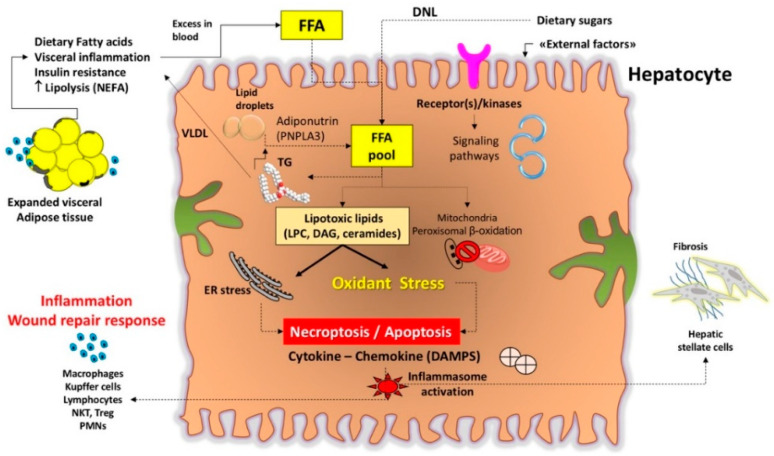
Mechanisms of lipotoxicity in the liver. Excessive accumulation of free fatty acids (FFA) is the result of increased influx, increased synthesis, decreased mitochondrial oxidation of FFA, or decreased secretion/export of FFA. Adiponutrin (also known as patatin-like phospholipase domain-containing protein 3, PNLPA3) governs the lipolysis of lipid droplets with FA flowing back to the hepatocyte FFA pool. With excess FFA, their disposal through beta-oxidation or formation of triglyceride is insufficient and lipotoxic species form (Lysophosphatidylcholine, LPC; diacylglycerol, DAG; ceramides) that mediate endoplasmic reticulum (ER) stress, oxidant stress and activation of the inflammasome (the multiprotein cytoplasmic complex that responds to damage-associated molecular patterns (DAMPs), as part of the innate immunity response). External factors, poorly understood so far, may include dysregulation of cytokines and adipokines, ATP depletion, toxic uric acid, sleep apnea leading to periodic hypoxia, and products of the gut microbiome (tumor necrosis factor (TNFa), endogenous ethanol, and endotoxins like lipopolysaccharides (LPS)). Such events promote the phenotype of NASH, hepatocellular injury, inflammation, stellate cell activation, and progressive accumulation of excess extracellular matrix. Further target structures of damage are other intracellular organelles, the nucleus, receptors and signaling pathways. See also [10,41,72,73].

**Figure 4 jcm-09-02648-f004:**
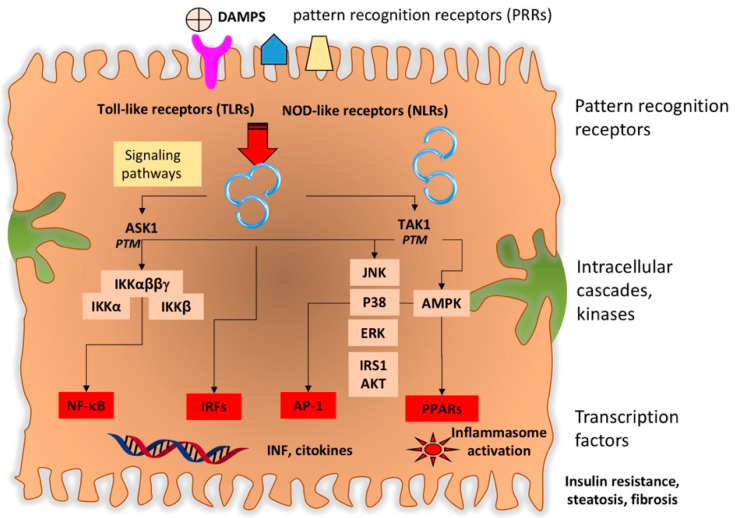
Detailed picture representing the ultimate consequence of circulating damage-associated molecular patterns (DAMs) on hepatocytes. DAMs represent the continuous stimulus for at least three intracellular events in the hepatocyte: (1) response of pattern recognition receptors (PRRs) such as Toll-like receptors (TLRs) and NOD-like receptors (NLRs). (2) Activation of intracellular cascades, kinases representing the downstream signaling pathways, such as the apoptosis signal-regulating kinase 1 (ASK1) and TGF-b-activated kinase 1 (TAK1). Both kinases are activated through post-transcriptional modification (PTM) and activate other key kinases (C-Jun N-terminal kinase, JNK), AMP-activated kinase, AMPK, and IkB). (3) Activation of transcription factors such as interferon regulatory factors (IRFs), nuclear factor (NF)-kB, activator protein 1 (AP-1), and peroxisome proliferator-activated receptors (PPARs). This ultimate event leads to the production of inflammatory cytokines and chemokines, with all metabolic consequences observed in NAFLD (insulin resistance, steatohepatitis, fibrogenesis, etc.). Additional endogenous targets regulating the innate immune elements in NASH include CASP8 and FADD-like apoptosis regulator (CFLAR), tumor necrosis factor (TNF) a-induced protein 3 (TNFAIP3), cylindromatosis (CYLD), transmembrane BAX inhibitor motif-containing 1 (TMBIM1), TNF receptor-associated factor 6 (TRAF6), TRAF1, TRAF3, dual-specificity phosphatase 14 (DUSP14), tripartite motif 8 (TRIM8), dickkopf-3 (DKK3), and TRAF5 (See also [41]).

**Figure 5 jcm-09-02648-f005:**
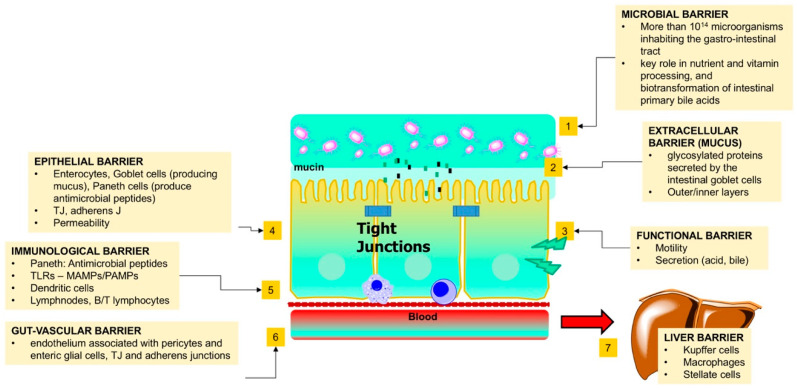
Components of the physiological gut barrier. The number of microorganisms inhabiting the gastro-intestinal tract has been estimated to exceed 10^14^. Starting from the gut lumen, we observe microbes (1) in the most superficial layer of the mucous (2). The functional barrier (3) includes the role of gastrointestinal motility ad secretion of gastric acid and bile the epithelial barrier (4) is composed by the collection of gut cells and tight junctions. The immunological barrier (5) includes antimicrobial peptides, toll-like receptors, microbial- (MAMPs) and pathogen- (PAMPs) associated molecular patterns, and B/T lymphocytes. The gut vascular barrier (6) and the ultimate liver barrier (7) are depicted, as well. Adapted from Nicoletti et al. [91].

**Figure 6 jcm-09-02648-f006:**
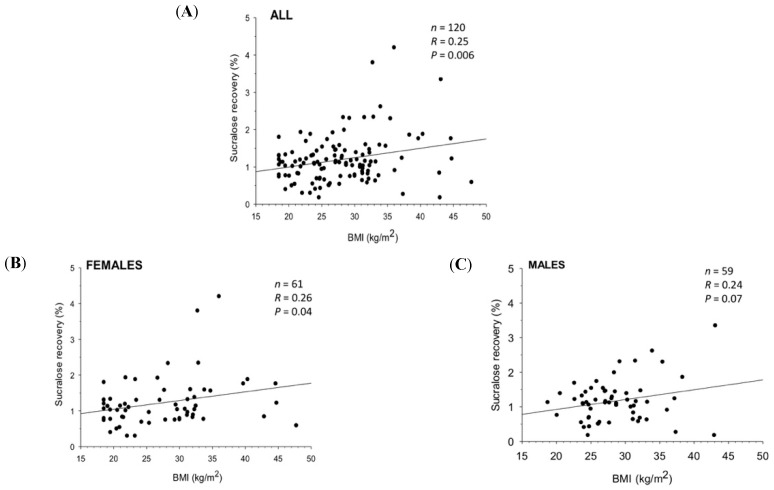
Colonic permeability (leakage) in a group of 120 subjects (61 females and 59 males; mean age 45 ± SEM1.2 years, range 18–75). Graphs show the correlation between urinary sucralose recovery (marker of colonic permeability) and body mass index (BMI) in: (**A**) all subjects; (**B**) females; (**C**) males. The prevalence of abnormal colonic permeability was higher in both overweight and obese than in normal weight subjects, and sucralose recovery increased significantly with BMI overall, in females (*p* = 0.04), and tended to increase also in males (*p* = 0.07). To measure overall intestinal permeability, four saccharide probes are administered orally and simultaneously. Stomach permeability is assessed by administering 20 g of sucrose; small intestine permeability is assessed by administering lactulose (5 g) and mannitol (1 g); and colonic permeability is assessed with 1 g of sucralose. A urinary sample is collected in the morning of the test, after an overnight fast. Sugars are dissolved in 250 mL of tap water and orally administered. Urines are collected every hour during six hours in a sterile container. Urinary samples are measured by chromatography/mass spectrometry (UPLC-MS/MS, AB Analitica, Padua, Italy). The fraction of the excreted sugars was calculated based on the quantity of sugar ingested by the patients and expressed as a percentage value, according to the manufacturer (AB analitica, Padua, Italy). Their presence in urine, in different amounts compared to the expected, indicated impaired permeability of the stomach, small intestine, or colon. Normal values for permeability test are expressed as a percentage) adapted from Di Palo et al. [167]).

**Table 1 jcm-09-02648-t001:** Most frequent causes of liver steatosis.

Metabolic: Nonalcoholic fatty liver (NAFLD) or metabolic-dysfunction-associated fatty liver disease (MAFLD)
Alcoholic liver disease (ALD)
Hepatitis C (in particular genotype 3)
Lipodystrophy
Parenteral nutrition
Starvation
Wilson disease
Abetalipoproteinemia
Drugs (e.g., methotrexate, tamoxifen, glucocorticoids, amiodarone, valproate, anti-retroviral agents for HIV)
Acute fatty liver of pregnancy
HELLP (hemolytic anemia, elevated liver enzymes, low platelet count) syndrome
Reye syndrome
Inborn errors of metabolism (LCAT deficiency, cholesterol ester storage disease, Wolman disease)
Drug-induced liver disease
Food contaminants of environmental origin (e.g., endocrine disrupting chemicals, persistent organic pollutants, metals)

**Table 2 jcm-09-02648-t002:** Lipid species accumulating in NAFLD.

**Free Fatty Acids (FFA)**
Originates from adipose tissue with insulin resistant status, from de novo lipogenesis (from carbohydrates), luminal nutrients, decreased export as TG in very-low density lipoproteins (VLDL) Internalized by liver plasma membrane transporter CD36 (increased expression during insulin resistant status) Especially toxic: saturated FFA (e.g., palmitate, stearate); less toxic: monounsaturated FFA (e.g., oleate)
**Triglycerides (TG)**
Originates from increased influx of FFAs in the liver and combination with one molecule of glycerol Accumulates as micro-, and macro-droplets At the early stage of NAFLD, TG represent a type of inert form protective against the ongoing lipotoxic injury [51,54]
**Lysophosphatidylcholine (LPC)**
Originates from phosphatidylcholine (intracellular action of phospholipase A2 or from extracellular lecithin-cholesterol acyltransferase) Can mediate intracellular damage (endoplasmic reticulum stress, activation of apoptotic pathways downstream of JNK), also interacting with FFA palmitate [55,56]
**Ceramides**
Originates from serine and palmitoyl-CoA (enzyme serine palmitoyltransferase) and from sphingomyelin (enzyme neutral sphingomyelinase) Can promote inflammation and cell toxicity via interaction with TNFα. Inhibition of ceramide synthesis decreases steatosis, cell injury, and insulin sensitivity in animal models of NAFLD [57] Mediates insulin resistance, cell toxicity, and pro-inflammatory effects (sequence IL-1 → ceramides → TNFα → inflammation) [57] Can induce release of extracellular vesicles (EV) → cell-cell communication (also in NASH)
**Free Cholesterol**
Can originate via the following pathway: liver steatosis/NASH → increased expression sterol regulatory element-binding protein (SREBP)-2 → upregulation of HMGCoA reductase → increased synthesis of free cholesterol (mitochondria) [58] → apoptosis → JNK-dependent pro-inflammatory pathways Role in inflammation, fibrosis, and liver injury [59] Target cells: hepatocytes, stellate cells, and Kupffer cells

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
