# Peer review of "Liver Steatosis, Gut-Liver Axis, Microbiome and Environmental Factors. A Never-Ending Bidirectional Cross-Talk"

_jcm, 2020, doi:10.3390/jcm9082648_

Round 1
Reviewer 1 Report
In the manuscript from Di Ciaula et al., described a very detailed review about liver steatosis and the gut liver axis as a bidirectional cross-talk.The review is very detailed and good described but the gut microbiota part contain several terminology mistakes and some concepts errors. The english writing need to revise. For instance, some sentences are very long and there are some punctuation mistakes that makes the reading less fluent. Furthermore, I suggest to the authors to avoid to write sentences as a list of words (example: lines 35-39, 234-236).
I report some examples below:
Line 31: change the word “ Mounting” with increasing
Lines 40: remove comma after the word PAMPs
Line 61-64: the sentence is too long. The authors should write 2 sentences instead of one.
Line 68: remove the comma after the word factor
Line 84: The author should explain the meaning of the sentence : “Butyrate, the short chain fatty acids, is protective while acetaldehyde can activate damage”.
Line 119: change the word features with characteristics
Lines 122-124: the sentence sounds not correct: “In a large Asian study in 16,279 non-obese subjects (BMI<25 122 kg/m2), patients displaying different grades of liver steatosis and fibrosis were at increased risk of 123 subclinical atherosclerosis. Please rewrite.
Line 124: Is there a mortality rate? If yes the authors should report it.
Line 138: rewrite the sentence as follow: ”The early fat storage in the liver could drive subclinical liver abnormalities, leading to NAFLD development and progression, that can increase risk for advanced liver disease and liver-related mortality.
Line 141: change the statement “ Great changes” with a more scientific word.
Lines 147-182: The acronyms need to be all defined in the text as well.
Line 147-150: accumulation is repeated two times. Use a synonymous.
line 162 : Rewrite the sentence : This step is also controlled by insulin and excess TG can be stored as lipid droplets or exported to blood as very-low-density lipoprotein .
line 214: change the word “of note” with notably.
line 229: change the verb join with a more scientific word.
Line 236: which areas of the brain?
Line 275: What is the meaning of “a general umbrella”?
Line 280: “floating” wrong terminology. The bacteria in the gut don’t float. Remove the verb floating in all the text.
Line 282: change the word reflections with consequences
Line 300, Figure 5 : wrong terminology:
The bacteria in the gut cannot be refer as a collection of strains.
And “huge collection” is not scientifically correct.
A proper sentence can be: “The number of microorganisms inhabiting the gastro-intestinal tract has been estimated to exceed 1014”
Line 291: “greatly manipulated” is not scientifically correct. Please rewrite.
Line 294-296: the sentence sound not correct. “The gut barrier integrity results from the ongoing equilibrium and cross-talk involving the microbiome, the mucus, the intestinal cells, the immunological system, and the gut-vascular barrier, and to maintenance of intestinal homeostasis” Please rewrite .
Line 292: “orthodox functions” wrong terminology. The authors need to clarify what they mean scientifically.
Line 318: wrong terminology “linen?”
Line 322: wrong terminology “postbiotics”. Secondary metabolites is the right term.
Line 320-321: It’s during a disease condition that bacteria might cross the mucus but this doesn’t happen in healthy subjects.
Line 333: Akkermansia muciniphila is a mucus degrading bacteria and its abundance is higher close to the mucus layer. https://doi.org/10.1016/j.bpg.2013.03.001
Line 341: change the sentence as follow: “Notably, the inner mucus layer is sterile….
Line 487: what is the meaning of “engagement”?
Line 562: change the word “insult” with a more scientific term.
Line 599: wrong terminology “powerful”. Please delete the word powerful
Line 399: Change the verb must with can or may.
Line 608: “blood supply” is not scientifically correct.
Line 645: “ vegetal carbohydrate”, the correct term is indigestible plant polysaccharides.
Line 869-870: remove the word “namely”. The phyla are well known in literature.
Line 893-896: the sentence is too long. Please abbreviate.
Line 901-902: This is still in debate if the changes start before in the microbiome or from the diet.
Line 903: wrong terminology :“clue”
Lines 901-907: the sentence sounds not right. Please rewrite.
Line 1011: what is the meaning of “bacterial manipulation” in the sentence.
Line 1013: “permeate with flow” is not scientifically correct.
Line 1021-1023: please rewrite the sentence for clarity.
Author Response
Point-to-point answers to reviewers
Reviewer #1. In the manuscript from Di Ciaula et al., described a very detailed review about liver steatosis and the gut liver axis as a bidirectional cross-talk.The review is very detailed and good described but the gut microbiota part contain several terminology mistakes and some concepts errors. The english writing need to revise. For instance, some sentences are very long and there are some punctuation mistakes that makes the reading less fluent. Furthermore, I suggest to the authors to avoid to write sentences as a list of words (example: lines 35-39, 234-236).
We thank the reviewer for the helpful comments, which certainly contributed to improve the manuscript. The English writing has been revised throughout the manuscript.
I report some examples below:
- Line 31: change the word “Mounting” with increasing
The text has been changed as suggested.
- Lines 40: remove comma after the word PAMPs
The text has been changed as suggested.
- Line 61-64: the sentence is too long. The authors should write 2 sentences instead of one.
The sentence has been changed as follow:
“Early identification and targeted treatment of patients with NAFLD will prevent consequences related to complications (including management of end-stage disease and HCC) and rising health cost. Further beneficial effects can derive from the reduction of risk factors for extrahepatic complications, which include cardiovascular disease and malignancy [9,10]”
- Line 68: remove the comma after the word factor
The text has been changed as suggested.
- Line 84: The author should explain the meaning of the sentence: “Butyrate, the short chain fatty acids, is protective while acetaldehyde can activate damage”.
The sentence has been changed as follow:
“Butyrate, the short chain fatty acids, is protective (i.e. improves the colonic defensive border), while acetaldehyde can activate a barrier damage”.
- Line 119: change the word features with characteristics
The text has been changed as suggested.
- Lines 122-124: the sentence sounds not correct: “In a large Asian study in 16,279 non-obese subjects (BMI<25 122 kg/m2), patients displaying different grades of liver steatosis and fibrosis were at increased risk of 123 subclinical atherosclerosis. Please rewrite.
The sentence has been changed as follow:
“In a large cross-sectional Asian study, the presence of NAFLD or AFLD was associated, also in non-obese subjects (BMI<25 kg/m2), with the score of coronary artery calcification, an expression of subclinical atherosclerosis [32].”
- Line 124: Is there a mortality rate? If yes the authors should report it.
The sentence “Lean subjects with NAFLD are also at higher risk of mortality [30]” has been now deleted from the manuscript.
- Line 138: rewrite the sentence as follow: ”The early fat storage in the liver could drive subclinical liver abnormalities, leading to NAFLD development and progression, that can increase risk for advanced liver disease and liver-related mortality.
We thank the reviewer for this comment. The sentence has been now rewritten as suggested.
- Line 141: change the statement “ Great changes” with a more scientific word.
The sentence has been changed as follow:
“A marked worsening occurs with advanced liver diseases, such as liver cirrhosis [37,38]”.
- Lines 147-182: The acronyms need to be all defined in the text as well.
All the acronyms have been now defined in the text.
- Line 147-150: accumulation is repeated two times. Use a synonymous.
The sentence has been changed as follow:
“NAFLD is due to excessive (>5%) storage of hepatic triglycerides (TG) as micro-, macro-vesicular deposits, associated with accumulation of free fatty acids (FFA), ceramides, as well as cholesterol [44,45]”.
- line 162 : Rewrite the sentence : This step is also controlled by insulin and excess TG can be stored as lipid droplets or exported to blood as very-low-density lipoprotein .
We thank the reviewer for this comment. The sentence has been now rewritten as suggested.
- line 214: change the word “of note” with notably.
We thank the reviewer. The word has been changed as suggested.
- line 229: change the verb join with a more scientific word.
The sentence has been changed as follow:
“The innate immune system contributes to metabolic inflammation with the recruitment of Kupffer cells, dendritic cells, lymphocytes, as well as hepatocytes and endothelial cells [66,67]”.
- Line 236: which areas of the brain?
The sentence has been changed as follow:
“In addition, the adipose tissue, skeletal muscle, the heart, the pancreatic islets, certain areas of the brain (mainly hippocampus, cerebellum, hypothalamus), and the intestinal microbiota represent additional organs potential targets of lipotoxicity [41,69-71].
- Line 275: What is the meaning of “a general umbrella”?
The terms “general umbrella” have been now deleted from the text. The sentence has been rewritten as follow:
“NAFLD develops because of the interaction of genes (epistasis) and environmental factors (exposome) [72,73]. The environmental factors act through intestinal, microbial, and dietary modifications [11,76], and can be linked with exposure to food contaminants, contaminated consumer products or air pollution [77-84].
- Line 280: “floating” wrong terminology. The bacteria in the gut don’t float. Remove the verb floating in all the text.
We thank the reviewer for this observation. The sentence has been changed as follow:
“For example, a diet enriched in fat could easily change the intestinal mucus [76] and the microbiome, leading to…”
Furthermore, the verb “floating” has been removed throughout the text,
- Line 282: change the word reflections with consequences
We thank the reviewer. The word has been changed as suggested.
- Line 300, Figure 5 : wrong terminology: The bacteria in the gut cannot be refer as a collection of strains. And “huge collection” is not scientifically correct.
A proper sentence can be: “The number of microorganisms inhabiting the gastro-intestinal tract has been estimated to exceed 1014”
The text (lines 302-303) has been changed as follow:
“The number of microorganisms inhabiting the gastro-intestinal tract has been estimated to exceed 1014. Starting from the gut lumen, we observe microbes (1) in the most…”
The text in the figure has been modified accordingly.
- Line 291: “greatly manipulated” is not scientifically correct. Please rewrite.
The sentence has been rewritten as follow:
“Moreover, BAs are markedly influenced by the intestinal microbiome but also control the intestinal microbiome…”
- Line 294-296: the sentence sound not correct. “The gut barrier integrity results from the ongoing equilibrium and cross-talk involving the microbiome, the mucus, the intestinal cells, the immunological system, and the gut-vascular barrier, and to maintenance of intestinal homeostasis” Please rewrite .
The sentence has been rewritten as follow:
“The gut barrier integrity results from a balanced cross-talk between microbiome, mucus, intestinal cells, immunological system and gut-vascular barrier. The stability of this equilibrium is essential in maintaining the intestinal homeostasis”.
- Line 292: “orthodox functions” wrong terminology. The authors need to clarify what they mean scientifically.
The term “orthodox” has been deleted. The sentence has been rewritten as follow:
“…beside their classical physiologic functions which allow…”
- Line 318: wrong terminology “linen?”
The sentence has been changed as follow:
“The microbiota in health is physically separated from the intestinal epithelium by the mucus”
- Line 322: wrong terminology “postbiotics”. Secondary metabolites is the right term.
The sentence has been changed as follow:
“In general, the microbiota is able to cross-talk with the host via several metabolic products. Secondary metabolites include short chain fatty acids…”
the term “postbiotics” has been replaced with “secondary metabolites” throughout the text
- Line 320-321: It’s during a disease condition that bacteria might cross the mucus but this doesn’t happen in healthy subjects.
We thank the reviewer for this comment. The sentence has been now deleted from the text.
- Line 333: Akkermansia muciniphila is a mucus degrading bacteria and its abundance is higher close to the mucus layer. https://doi.org/10.1016/j.bpg.2013.03.001
We thank the reviewer for this indication. The sentence has been changed as follow:
“In addition, some bacteria use mucus for nutrition and modulate inflammatory changes. In particular, Akkermansia muciniphila is a mucus degrading bacteria and its abundance is higher close to the mucus layer [109]. This anaerobe, Gram negative, mucus degrading specialist populates the intestinal lumen [110,111] and its reduced abundance is associated with inflammation and impaired barrier integrity, and non-alcoholic liver damage [112,113].”
- Line 341: change the sentence as follow: “Notably, the inner mucus layer is sterile….
The text has been changed as suggested.
- Line 487: what is the meaning of “engagement”?
The term “engagement” has been deleted. The sentence has been changed as follow:
“or by engaging activating NK cell receptors expressed by intestinal intraepithelial lymphocytes lineages [160].”
- Line 562: change the word “insult” with a more scientific term.
The sentence has been changed as follow:
“and detrimental agents perpetuate the local damage in the liver but could also activate a systemic inflammatory response”
- Line 599: wrong terminology “powerful”. Please delete the word powerful
- Line 599: Change the verb must with can or may.
The words “powerful” and “must” has been deleted. The sentence has been rewritten as follow:
“Gut microbiota, a super-organism hosted in the human body, can be involved in the pathogenesis of NAFLD [75]. …”
- Line 608: “blood supply” is not scientifically correct.
The sentence has been changed as follow:
“The majority of blood flow reaches the liver…”
- Line 645: “ vegetal carbohydrate”, the correct term is indigestible plant polysaccharides.
We thank the reviewer for his suggestion. The sentence has been rewritten as follow:
“By contrast, a diet rich in fibre and indigestible plant polysaccharides increases Prevotella abundance.”
- Line 869-870: remove the word “namely”. The phyla are well known in literature.
The sentence has been changed as follow:
“3-(4-hydroxyphenyl) lactate correlated with specific bacterial species (Firmicutes, Bacteroidetes, and Proteobacteria), as well as hepatic fibrosis [259].”
- Line 893-896: the sentence is too long. Please abbreviate.
The sentence has been changed as follow:
Conversely, protective effects and a possible reversal of liver fat accumulation and lipotoxicity could be driven by favourable environmental factors modulating composition and relative abundance of specific phylotypes.
- Line 901-902: This is still in debate if the changes start before in the microbiome or from the diet.
The sentence has been changed as follow:
“Changes are likely due to the initial modification of the microbiota, rather than the dietary pattern per se [77]. This topic, however, is still under discussion.”
- Line 903: wrong terminology :“clue”
- Lines 901-907: the sentence sounds not right. Please rewrite.
The sentence has been changed as follow:
“A Western-style diet is rich in saturated fat and sugars, and might promote an injury. In mice, a high-fat diet or fiber-deprived diets change the composition of intestinal microbiota and can damage the intestinal barrier through increased intestinal permeability, reduced thickness of the mucous layer, abnormalities of tight junction proteins of the epithelial barrier and low-grade intestinal inflammation[104,370,371]”
- Line 1011: what is the meaning of “bacterial manipulation” in the sentence.
The words “bacterial manipulation” have been deleted. The sentence has been changed as follow:
“Complex interactions involve bacteria and intestinal barrier, which come in close contact with dietary factors, food contaminants, drugs, chemicals, bile,…”
- Line 1013: “permeate with flow” is not scientifically correct.
The sentence has been changed as follow:
“Complex interactions involve bacteria and intestinal barrier, which come in close contact with dietary factors, food contaminants, drugs, chemicals, bile. All these elements can modulate the pathways linking the gut with the liver, leading to both local and systemic effects”
- Line 1021-1023: please rewrite the sentence for clarity.
The sentence has been rewritten as follow:
According to available evidence, the majority of the environmental factors leading to abnormalities in the gut-liver axis and, in turn, to liver and metabolic dysfunction could be managed. Future studies should therefore target the modulation of gut-liver homeostasis with both preventive and therapeutic goals.

Reviewer 2 Report
In this manuscript, the authors summarized information related to the association between NAFLD and the physical changes in the gut. Gut-liver axis has been extensively studied and a large amount of evidence shows that the gut has significant impacts on the pathogenesis of various liver diseases. This review manuscript discussed the current knowledge about NAFLD, gut-liver axis, and how gut can affect the progress of NAFLD. This is an interesting review and shows a large amount of evidence. There are some small points which may help the authors to improve the manuscript.
1. The title should be changed. What does "environment" indicate? This manuscript did not show anything regarding to environmental changes.
2. The authors should use more descriptive titles for each section. It is very difficult to point out the logic of this review based on titles of each section.
3. Line 153-181 should be re-organized.
Author Response
Point-to-point answers to reviewers
Reviewer #2
In this manuscript, the authors summarized information related to the association between NAFLD and the physical changes in the gut. Gut-liver axis has been extensively studied and a large amount of evidence shows that the gut has significant impacts on the pathogenesis of various liver diseases. This review manuscript discussed the current knowledge about NAFLD, gut-liver axis, and how gut can affect the progress of NAFLD. This is an interesting review and shows a large amount of evidence. There are some small points which may help the authors to improve the manuscript.
We thank the reviewer for the interest and for the helpful comments.
- The title should be changed. What does "environment" indicate? This manuscript did not show anything regarding to environmental changes.
The title has been changed as follow: “LIVER STEATOSIS, GUT-LIVER AXIS, MICROBIOME AND ENVIRONMENTAL FACTORS. A NEVER-ENDING BIDIRECTIONAL CROSS-TALK”.
In particular, environmental factors able to modulate/affect the gut-liver axis (also promoting NAFLD) have been discussed:
- in table 1
- in section 3 (“Beyond NAFLD: the gut liver-axis, the gut barrier and the liver barrier”)
- in section 4.2 (“Oxidative stress”)
- more extensively, in section 5 (“The microbiota as a target of environmental factors involved in NAFLD”)
- The authors should use more descriptive titles for each section. It is very difficult to point out the logic of this review based on titles of each section.
We thank the reviewer for this constructive comment.
The sections of the manuscript have been now re-titled as follow:
- Introduction
- Risk factors, clinical features, and pathophysiology of NAFLD
- Beyond NAFLD: the gut liver-axis, the gut barrier and the liver barrier
- The main constituents of the ongoing liver damage
- The microbiota as a target of environmental factors involved in NAFLD
- Conclusions
- Line 153-181 should be re-organized.
Lines 153-181 have been now re-organized as follow:
“In health, as shown in Figure 2, there are three essential steps of FFA accumulation in the liver:
- Influx of non-esterified fatty acids (NEFA). This step accounts for ~60% of total FFAs in the liver and brings to enrichment of the FFA pool in hepatocytes. NEFA originate from lipolysis of triglycerides in adipocyte under the control of insulin [46]. Once in the liver, FFAs have different fates:
- as fatty acyl-CoA, FFAs can enter the mitochondria under the control of carnitine palmitoyltransferase (CPT)-1 and undergo β-oxidation to acetyl-CoA joining the tricarbossilic acid cycle with production of ATP;
- FFAs are also esterified to TG (no more than 5% in the hepatocyte) via the key enzymes diaglyceride acyltransferase (DGAT)1 and DGAT2. This step is also controlled by insulin and excess TG can be stored as lipid droplets or exported to blood as very-low-density lipoprotein. [47].
- TG can also be hydrolyzed under the actions of hydrolases such as the Patatin-like phospholipase domain-containing protein 3 (PNPLA3), also known as adiponutrin, to enrich the fatty acid (FA) pool [46,48].
- De novo lipogenesis (DNL) of FAs. This step accounts for ~25% of total FFAs in the liver and originate from dietary sugars. With DNL, hepatocytes convert excess glucose and fructose to FAs. Insulin mediates the transport of absorbed dietary carbohydrates to its target tissues (skeletal muscle and liver for storage as glycogen). Glucose in hepatocytes is partly metabolized to pyruvic acid via glycolysis and then to acetyl-CoA, to generate ATP in the tricarboxylic acid cycle and promote gluconeogenesis during hypoglycaemia. The first step of DNL is the synthesis of malonyl-CoA from cytosolic acetyl-CoA by acetyl-CoA carboxylase (ACC). Malonyl-CoA serves as a substrate to form saturated FA, by FA synthase (FAS). Stearoyl CoA desaturase (SCD)1 is an endoplasmic reticulum (ER) enzyme that is responsible for the formation of monounsaturated FAs from saturated FAs [41].
- Influx of dietary FAs. Influx of FAs from diet is ~15% of all amount of FFAs in the liver [48]. Bile acids (BA) hydrolyse intestinal dietary TGs to form nascent chylomicrons. The FAs made from the hydrolysis of TGs, are taken up by adipose tissue and liver [49]. At the same time, BA act as potent metabolic regulators in the terminal ileum by activating the farnesoid X receptor (FXR) plus pregnane X receptor (PXR), and the G-protein-coupled bile acid receptor-1 (GPBAR-1) with effects in the liver and in the muscle, adipocytes and brown adipose tissue for energy expenditure [12,50].”

Round 2
Reviewer 1 Report
The authors improved the manuscript , the English language is now readable and doesn’t need further changes. The terminology is now correct.